# Optimization of surface roughness for titanium alloy based on multi-strategy fusion snake algorithm

**Nanqi Li[1], ZuEn Shang[1]\*, Yang Zhao[2], Hui Wang[1], Qiyuan Min[1]**

**1** School of Mechanical Engineering, Liaoning Technical University, Fuxin, China, **2** School of Electronic and Information, Liaoning Technical University, Fuxin, China

\* shangzuen@126.com

## Abstract

Titanium alloy is known for its low thermal conductivity, small elastic modulus, and propensity for work hardening, posing challenges in predicting surface quality post high-speed milling. Since surface quality significantly influences wear resistance, fatigue strength, and corrosion resistance of parts, optimizing milling parameters becomes crucial for enhancing service performance. This paper proposes a milling parameter optimization method utilizing the snake algorithm with multi-strategy fusion to improve surface quality. The optimization objective is surface roughness. Initially, a prediction model for titanium alloy milling surface roughness is established using the response surface method to ensure continuous prediction. Subsequently, the snake algorithm with multi-strategy fusion is introduced. Population initialization employs an orthogonal matrix strategy, enhancing population diversity and distribution. A dynamic adaptive mechanism replaces the original static mechanism for optimizing food quantity and temperature, accelerating convergence. Joint reverse strategy aids in selecting and generating individuals with higher fitness, fortifying the algorithm against local optima. Experimental results across five benchmarks employing various optimization algorithms demonstrate the superiority of the MSSO algorithm in convergence speed and accuracy. Finally, the multi-strategy snake algorithm optimizes the objective equation, with milling parameter experiments revealing a 55.7 percent increase in surface roughness of Ti64 compared to pre-optimization levels. This highlights the effectiveness of the proposed method in enhancing surface quality.

## 1 Introduction

In recent years, to meet the service requirements of high-end equipment under extreme working conditions, titanium alloy is used in the selection of a large number of structural parts [1]. Because titanium alloy has low density, high-temperature resistance, wear resistance, corrosion resistance, impact resistance, and other excellent mechanical properties, it is widely used in aerospace, deep sea exploration, biomedical, and other fields [2]. Surface finish plays a crucial role in the service life of parts, which ensures the reliability of sensitive aerospace and other

**Data Availability Statement:** All relevant data are within the article.

**Funding:** This study was supported by the project of Liaoning Provincial Science and Technology Department (2023-BS-204). The role of funders

Funding acquisition, Project administration and Resources ZuEn Shang.

**Competing interests:** The authors have declared that no competing interests exist.

**Abbreviations:** $X_1$, $X_2$, $X_3$, $X_4$, Milling parameters: spindle speed, width of cut, depth of cut, feed per tooth; $Ra$, surface roughness; $\beta_0$, $\beta_s$, $\beta_{ss}$, $\beta_{sd}$, Regression coefficients: constant term, primary term coefficient, squared term coefficient, interaction term coefficient; $Q$, Total amount of food, snake algorithm parameters; $Temp$, Temperature parameters, snake algorithm control parameters; $gbest$, global optimum solution; $i,j$, Different factors or levels in a multifactor test; $m,f$, Individual male and female snakes; $t$, Number of iterations or time steps; $R^2$, coefficient of determination; Adjusted $R^2$, Adjusted coefficient of determination; Predicted $R^2$, Projected coefficient of determination.

fields. However, the difficult machining characteristics of titanium alloy limit its further application and the improvement of processing efficiency. The low thermal conductivity of titanium alloy makes the cutting zone temperature rise faster, and it is easy to produce chip welding and cutting shock. Therefore, academia and industry are committed to optimizing titanium alloy milling parameters to improve processing efficiency and processing quality. According to the research of Veiga, C [3], the difficult machining characteristics of titanium alloy are mainly reflected in the following aspects. First of all, due to the high chemical activity of titanium alloy, it is easy to chemically react with cutting tools, resulting in serious wear of cutting tools. The limits the improvement of cutting speed and cutting depth and reduces the processing efficiency and quality. This view is also supported by the research of Liu, H [4], who found that the chemical reaction between the cutting tool and the titanium alloy is one of the main reasons leading to the difficulty of the cutting process. Secondly, titanium alloy has a low thermal conductivity, which leads to the heat generated during the machining process not easy to dissipate quickly, thus heating the workpiece and the tool. This will cause cutting deformation, tool wear, and other problems, the machining accuracy and tool life have an adverse effect. In the study of Gupta, M [5], they pointed out that the low thermal conductivity of titanium alloy limited the thermal control of the cutting process, thus making the machining process more complicated and difficult. In addition, titanium alloys also exhibit strong solid solution curing. According to the research of A. Srivastava and M. Sharma [6], in the cutting process, titanium alloy is easy to form solid solution and solid solution compounds, so that the cutting material and the workpiece produce strong adhesion and welding. This can lead to serious tool wear and surface quality degradation. Therefore, in the titanium alloy cutting process, it is necessary to adopt the appropriate cutting parameters and tool structure to reduce the cutting force and improve the cutting surface quality.

With the development of computer technology, more and more work can be replaced by PC [7], and the optimization of cutting parameters is no exception. Many results show that the computing power of PC coupled with appropriate code can easily get the required results. Li, W et al. (2014) proposed a model based on deep reinforcement learning to optimize process parameters and machine tool energy consumption [8]. Taking the machining center as an example, the aluminum alloy workpiece is milled. Compared with the classical optimization algorithm, this method can save 95% of the optimization calculation time, and ensure that the average machining cost after optimization is close to the minimum machining cost obtained by the classical optimization algorithm. Through experimental verification, they proved the accuracy and reliability of the model. Chen, Y et al. (2018) studied the prediction and analysis of cutting forces during the high-speed milling of titanium alloys [9]. They adopted a hybrid method-based model to predict the magnitude of cutting force by inputting cutting parameters and material properties. Their results show that the model can accurately predict the cutting force and provide a way to guide the high-speed milling process. Prakash, C et al. (2015) reviewed the machining of titanium alloys and proposed a model based on a genetic algorithm to optimize cutting parameters [10]. Their research shows that the surface quality of titanium alloy cutting can be improved by selecting appropriate parameters such as cutting speed, feed speed and cutting depth.

At the same time, to optimize the milling parameters of titanium alloy, an advanced artificial intelligence algorithm, snake optimization algorithm in the swarm intelligence algorithm is adopted in this paper. Snake Optimization (SO) in swarm intelligence algorithm [11] is a new meta-heuristic algorithm proposed by Fatma A., Hashim et al. It has more abundant optimization methods in the iterative process and fewer adjustment parameters, but SO algorithm has a slow convergence speed in the early stage and poor global search ability. It is easy to converge to the local optimal solution, etc.

As technology continues to advance, optimisation problems in engineering design are becoming more and more complex, especially in fields such as aerospace, mechanical design, and structural engineering [12]. Multi-objective optimisation problems in these fields require multiple performance metrics to be satisfied while taking into account various factors such as cost, efficiency, and safety. Traditional optimisation methods are often difficult to handle these complex requirements simultaneously, and metaheuristic algorithms, with their global search capability and flexibility, show great potential in solving such problems [13,14]. These studies are of great importance in the field of engineering design.

Aiming at the existing problems of the SO algorithm, the relevant improvement research is carried out according to the existing literature. The improved algorithm in [15–17] uses levy flight strategy to increase the diversity of particles in the later iteration period, effectively improving the optimization ability of the algorithm. The improved algorithm in [18,19] uses the dynamic reverse strategy to solve the problem of premature convergence and convergence to the local optimal. A population initialization method based on orthogonal array is proposed to speed up the early convergence of the algorithm [20]. A quasi-physical strategy-based PSO algorithm (QPS-PSO) is proposed [21], which improves the optimization model of the PSO algorithm, and triggers the specified convergence model according to fitness, and solves the problem of premature convergence and convergence to local optimal in solving multi-modal global optimization problems.

To sum up, to solve the difficult machining characteristics of titanium alloy mainly includes high chemical activity, low thermal conductivity and strong solid solution curing effect makes it difficult to obtain good surface roughness, scholars have proposed a variety of methods to improve the machining performance of titanium alloy, including a reasonable selection of cutting parameters, the use of advanced processing technology, the selection of appropriate tool materials and coatings and other measures. In addition, optimization of cutting parameters, the use of internal cooling tools and coolants and other measures can effectively control the temperature during titanium alloy processing, improve the processing effect, and reduce thermal damage. These studies provide an important theoretical and practical basis for the development of titanium alloy processing technology. These studies provide effective methods and guidance for parameter optimization in titanium alloy cutting process. The optimization of the titanium alloy cutting algorithm is studied deeply, and a series of improvement methods are put forward from many angles. Reasonable selection and optimization of cutting parameters can significantly improve surface roughness, extend tool life, improve cutting efficiency and machining quality. These research results provide important theoretical and practical guidance for titanium alloy cutting.

In this study, an optimization method of milling process parameters based on improved snake algorithm was proposed, and a prediction model of milling surface roughness of titanium alloy was established based on the response surface method considering different milling parameters. At the same time, an improved snake algorithm (MSSO) based on multi-strategy mixing was proposed. Firstly, the orthogonal array was introduced to initialize the population, which enlarges the global exploration space of the initial population and increases the diversity of the initial population, thus improving the convergence speed and precision of the algorithm. Secondly, the exploration stage and development stage were dynamically selected according to the fitness of the algorithm in the optimization process, to effectively reduce the unnecessary time of exploration or developmentstage and accelerate the convergence speed, Finally, the joint reverse selection strategy was used to replace the population renewal method of the original algorithm. The new update strategy ensures the population diversity of the development stage enhances the global nature of the algorithm, and improves the ability of the algorithm to find the best solution. The method was applied to the optimization of milling parameters to

improve the surface roughness of titanium alloy. The experimental results show that the roughness optimization speed is faster for improving the surface roughness of titanium alloy.

## 2 Prediction model of titanium alloy surface roughness based on response surface method

### 2.1 High-speed milling experimental platform

The test specimen is an aviation casting titanium alloy TI64 used for the tail fin of an aircraft with a length of 100mm and a width of 60mm.The chemical composition of Ti64 is roughly: Aluminium (Al): 6.20% Vanadium (V): 3.82% Iron (Fe): 0.043% Carbon (C): 0.010% Nitrogen (N): 0.016% Hydrogen (H): 0.0002% Oxygen (O): 0.10%.These chemical compositions together determine the excellent properties of the Ti-6Al-4V alloy. The specimen is fixed on the workbench by special tooling, and the cutter ran along one end of the specimen according to the designed milling parameters [22]. The high-speed milling experiment site and surface quality test platform are shown in Fig 1.

The specific implementation conditions of the experiment are: high-speed vertical machining center (VMC2100-B), Mitsubishi high-speed coated doubleedged square pointed blade (APMT1604PDER-h2). The milling methods are face milling, up and down milling.The Keyence 5000 ultra depth of field 3D micro system is used to test the milled surface of the specimen. The experiments in this paper used an oil-based coolant.Because there were multiple measurable points during the experiments it was not necessary to repeat the experiments multiple times, each experiment was repeated once, and in order to minimise the effects of tool wear the blades were replaced with new ones before the start of each set of experiments.

The 3D surface topography measurements mentioned in this paper were made using a PS50 3D surface topographer, which uses a white light confocal technique for the measurements. In 3D measurements, a commonly used parameter is Sa (Arithmetic Mean Height), which is similar to Ra (Arithmetic Mean Roughness), but Sa is a parameter for 3D surface

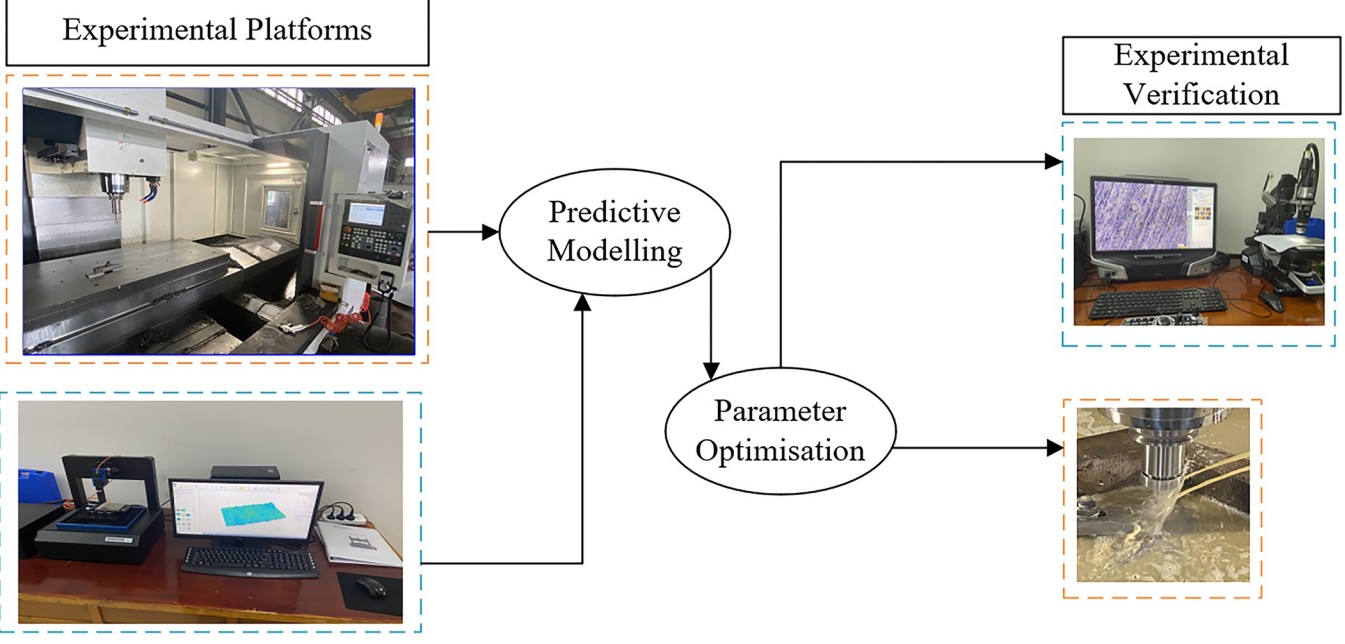

**Fig 1. Experimental flow chart.**

measurements, whereas Ra is usually used for 2D measurements.Sa is calculated by summing up the absolute value of the deviation of the heights of all the measured points from their mean heights and dividing by the absolute value of the deviation of the heights of all the measured points from their mean heights. Sa is calculated by summing the absolute value of the deviation of the heights of all the measured points from their average height and dividing by the total number of measured points.Sa provides an overall measure of the surface roughness, including both height and magnitude information.

We used advanced measurement techniques to evaluate the surface roughness of titanium alloy materials after high-speed milling. Specifically, a PS50 3D surface profiler was used to perform precision surface roughness measurements. This device is based on white light confocal technology, which emits a specific wavelength of light and receives the reflected light back through the probe. By calculating the distance between the probe and the surface, the device is able to accurately acquire the 3D coordinates of the surface.The collected data is analysed in detail using the NANOVIA 3D software to obtain surface roughness measurements from the nanometre to the millimetre scale. The advantages of this measurement method are its high accuracy, fast measurement and good repeatability, providing us with reliable surface roughness data. In addition, in order to further observe the macroscopic morphology of the machined surface, we also used a Japanese Keens VHX-5000 super depth-of-field microscope. This microscope provides a clear image of the surface, helping us to understand the distribution of surface roughness and morphological details more intuitively.By combining these two measurement techniques, we are not only able to obtain quantitative data on surface roughness, but also to conduct in-depth analysis of its micro and macro features, which is of great significance for optimising milling process parameters and improving machining quality.

## 2.2 Experimental test based on response surface method

In order to explore the influence law of high-speed milling parameters and their interaction effects on the surface roughness of aero-cast titanium alloy TI64 and seek the best parameter matching, the experiments were designed based on Response Surface Methodology (RSM). RSM is a nonlinear modeling analysis method based on experimental conditions. In this experiment, the milling surface roughness of aero-cast titanium alloy TI64 specimen was taken as the evaluation index, and the Box-Behnken design experimental design method in RSM was selected to arrange the high-speed milling experimental scheme. The four-factor and three-level coding table of milling parameters is shown in Table 1, and the experimental results of RSM are shown in Table 2.

## 2.3 Determination of response surface analysis model

Design-Expert provides a variety of analysis models between variance and determination coefficient to determine the degree of fitting between regression coefficient and experimental regression model. This paper obtains the ANOVA relationship between milling parameters and surface roughness and the analysis relationship of multiple prediction coefficients of response value through experiments, as shown in Tables 3–5.

**Table 1. Horizontal coding of milling parameters.**

| level | factor | | | |
|---|---|---|---|---|
| | Spindle speed:$X_1$/rpm | Cutting width:$X_2$/mm | Cutting depth:$X_3$/mm | Feed per tooth:$X_4$/mm |
| -1 | 500 | 2 | 0.1 | 0.01 |
| 0 | 1500 | 5 | 0.25 | 0.03 |
| 1 | 2500 | 8 | 0.4 | 0.04 |

**Table 2. Experimental program and test results.**

| Serial number | $X_1$ (rpm) | $X_2$ (mm) | $X_3$ (mm) | $X_4$ (mm) | Ra (μm) |
|---|---|---|---|---|---|
| 1 | 500 | 5 | 0.4 | 0.03 | 0.566 |
| 2 | 1500 | 5 | 0.25 | 0.03 | 0.412 |
| 3 | 2500 | 2 | 0.25 | 0.03 | 0.283 |
| 4 | 1500 | 2 | 0.25 | 0.01 | 0.496 |
| 5 | 1500 | 5 | 0.25 | 0.03 | 0.381 |
| 6 | 2500 | 5 | 0.25 | 0.04 | 0.297 |
| 7 | 2500 | 5 | 0.4 | 0.03 | 0.608 |
| 8 | 500 | 5 | 0.25 | 0.01 | 0.603 |
| 9 | 1500 | 2 | 0.1 | 0.03 | 0.207 |
| 10 | 500 | 5 | 0.1 | 0.03 | 0.234 |
| 11 | 1500 | 5 | 0.25 | 0.03 | 0.344 |
| 12 | 2500 | 5 | 0.25 | 0.01 | 0.637 |
| 13 | 1500 | 5 | 0.4 | 0.01 | 0.763 |
| 14 | 1500 | 5 | 0.1 | 0.04 | 0.215 |
| 15 | 1500 | 8 | 0.25 | 0.01 | 0.675 |
| 16 | 2500 | 8 | 0.25 | 0.03 | 0.232 |
| 17 | 1500 | 2 | 0.4 | 0.03 | 0.443 |
| 18 | 1500 | 8 | 0.1 | 0.03 | 0.328 |
| 19 | 500 | 5 | 0.25 | 0.04 | 0.264 |
| 20 | 500 | 8 | 0.25 | 0.03 | 0.511 |
| 21 | 1500 | 2 | 0.25 | 0.04 | 0.224 |
| 22 | 500 | 2 | 0.25 | 0.03 | 0.254 |
| 23 | 1500 | 5 | 0.4 | 0.04 | 0.466 |
| 24 | 1500 | 5 | 0.25 | 0.03 | 0.382 |
| 25 | 1500 | 5 | 0.1 | 0.01 | 0.434 |
| 26 | 2500 | 5 | 0.1 | 0.03 | 0.253 |
| 27 | 1500 | 8 | 0.25 | 0.04 | 0.404 |
| 28 | 1500 | 5 | 0.25 | 0.03 | 0.397 |
| 29 | 1500 | 8 | 0.4 | 0.03 | 0.664 |

According to the variance and determination coefficient analysis in Tables 3–5, in the quadratic model of surface roughness as response value, $R^2$ and adjusted $R^2$ are close to 0.9, which proves that the correlation between the model and the experiment is high and the model is relatively accurate. The predicted $R^2$ is close to 0.8, and the predicted residual sum of squares is the smallest, which proves that the prediction and universality of the model are strong, which indicates that the quadratic model is highly reliable when analyzing the surface roughness. Therefore, the quadratic model is chosen as the analysis model of surface roughness.

**Table 3. Comprehensive analysis model.**

| Category of model | P | Adjusted $R^2$ | Predict $R^2$ | Result |
|---|---|---|---|---|
| Linear model | <0.0001 | 0.8318 | 0.7808 | - |
| 2FImodel | 0.1676 | 0.8583 | 0.7280 | - |
| Quadratic model | <0.0001 | 0.9683 | 0.9296 | Suggestions |
| Cubic model | 0.9292 | 0.9483 | 0.1111 | Ignore |

**Table 4. Analysis of variance model.**

| Source | Sum of squares | df | Mean square | F value | P value | Result |
|---|---|---|---|---|---|---|
| Mean model | 5.07 | 1 | 5.07 | - | - | - |
| Linear model | 0.53 | 4 | 0.13 | 35.62 | <0.0001 | - |
| 2FImodel | 0.033 | 6 | 5.517E-003 | 1.75 | 0.1676 | - |
| Quadratic model | 0.047 | 4 | 0.012 | 16.63 | <0.0001 | Suggestions |
| Cubic model | 2.970E-003 | 8 | 3.713E-004 | 0.32 | 0.9292 | Ignore |
| Pure Error | 6.918E-003 | 6 | 1.153E-003 | - | - | - |
| Cor Total | 5.69 | 29 | 0.20 | - | - | - |

The linear mode has an $F$-value of 35.62 and a $P$-value of 0.0001, which is much less than 0.05, indicating that at least one of the milling parameters in the model has a significant main effect on $Ra$. Also spindle speed ($X_1$), width of cut ($X_2$), depth of cut ($X_3$) and feed per tooth ($X_4$) as components of the linear model have a direct effect on Ra. Since the p-values are not given, we cannot determine the specific significance of each parameter, but the significance of the overall linear model suggests that these parameters, at least in general, have a significant effect on $Ra$. From the interaction effect analysis, the $F$-value regarding the quadratic model was 16.63 with a $P$-value of 0.0001, indicating that the interaction effect between the milling parameters also had a significant effect on $Ra$. This may include interactions between parameters such as $X_1X_2$, $X_1X_3$, $X_1X_4$, $X_2X_3$, $X_2X_4$ and $X_3X_4$. The contribution can be assessed by the $R^2$ value of the model, which indicates the amount of variance explained by the model. Comparing the results of similar studies in the literature can help us understand the milling characteristics of the titanium alloy Ti64. Other studies may have reported $Ra$ values for different materials or different cutting conditions, and these data can be used to compare and validate the findings of this study. The effect of milling parameters on $Ra$, from the machinability point of view, the milling parameters listed as cutting speed, feed rate and depth of cut directly affect the material removal rate and heat generation, which in turn affects the surface roughness. Higher cutting speeds may lead to higher cutting temperatures, which in turn affects the plastic deformation of the material and $Ra$. It is also recognised that statistical methods, although capable of revealing the effect of milling parameters on $Ra$, may not be able to fully account for all the observed phenomena. Factors such as the microstructure of the material, tool wear, and the cooling effect of the cutting fluid may have an important effect on $Ra$, but these factors may not be adequately represented in the statistical models. Regarding additional contributions in order to provide deeper insights, studies could combine experimental results and theoretical analyses to explore how milling parameters can change $Ra$ by affecting the material removal mechanism.

## 2.4 Regression model

Response surface method is a combination of mathematical methods and statistical methods, using regression method as a equation estimation tool, the relationship between the factors in

**Table 5. Analysis model of determinable coefficient.**

| Source | S | $R^2$ | Adjusted $R^2$ | Predict $R^2$ | SSE | Result |
|---|---|---|---|---|---|---|
| Linear model | 0.061 | 0.8558 | 0.8318 | 0.7808 | 0.14 | - |
| 2FImodel | 0.056 | 0.9089 | 0.8583 | 0.7280 | 0.17 | - |
| Quadratic model | 0.027 | 0.9842 | 0.9683 | 0.9296 | 0.044 | Suggestions |
| Cubic model | 0.034 | 0.9889 | 0.9483 | 0.1111 | 0.55 | Ignore |

the multi-factor test and the test results (response value) is approximated by polynomial, the relationship between the factors and the test results is equationally analyzed quantitatively the influence of each factor and their interaction on the response value, the ultimate goal is to optimize the response value. Its second-order polynomial model is as follows:

$$y = \beta_0 + \sum_{s=1}^{u} \beta_s x_s + \sum_{s=1}^{u} \beta_{ss} x_s^2 + \sum_{s<d}^{u} \beta_{sd} x_s x_d \tag{1}$$

Where: $u$ is the number of design variables, $x$ is the $s$ and $d$ design variables, $\beta_0, \beta_s, \beta_{ss}, \beta_{sd}$ are regression coefficients, which can be determined by regression analysis.

The influence of milling parameters on surface roughness is not only single factor, the interaction of two factors on optimization can not be ignored, considering the interaction effect and secondary effect between the factors, the four-factor second-order regression model expression is as follows:

$$\begin{aligned}
y = \beta_0 + \beta_1 x_1 + \beta_2 x_2 + \beta_3 x_3 + \beta_4 x_4 + \beta_{11} x_1^2 + \\
\beta_{22} x_2^2 + \beta_{33} x_3^2 + \beta_{44} x_4^2 + \beta_{12} x_1 x_2 + \beta_{13} x_1 x_3 + \\
\beta_{14} x_1 x_4 + \beta_{23} x_2 x_3 + \beta_{24} x_2 x_4 + \beta_{34} x_3 x_4
\end{aligned} \tag{2}$$

Considering the interactive and secondary effects of milling parameters, the coefficient terms of the regression equation are solved according to the coded data in the table, and the coded terms can be centralized processed by Eq (3):

$$x'_{sd} = x_{sd}^2 - \frac{1}{N} \sum_{d=1}^{N} x_{sd}^2 \tag{3}$$

Then the coefficient of the constant term in the regression equation can be expressed as:

$$\beta_0 = \frac{1}{n} \sum_{d=1}^{29} y_d = \bar{y} \tag{4}$$

The other coefficients are:

$$\begin{cases}
\beta_1 = \dfrac{\sum\limits_{d=1}^{29} x_{1d} y_d}{\sum\limits_{d=1}^{29} x_{1d}^2}, \beta_2 = \dfrac{\sum\limits_{d=1}^{29} x_{2d} y_d}{\sum\limits_{d=1}^{29} x_{2d}^2}, \beta_3 = \dfrac{\sum\limits_{d=1}^{29} x_{3d} y_d}{\sum\limits_{d=1}^{29} x_{3d}^2} \\[4mm]
\beta_{11} = \dfrac{\sum\limits_{d=1}^{29} (x'_{1d}) y_d}{\sum\limits_{d=1}^{29} (x'_{1d})^2}, \beta_{12} = \dfrac{\sum\limits_{d=1}^{29} (x_1 x_2)_d y_d}{\sum\limits_{d=1}^{29} (x_1 x_2)_d^2}, \beta_{13} = \dfrac{\sum\limits_{d=1}^{29} (x_1 x_3)_d y_d}{\sum\limits_{d=1}^{29} (x_1 x_3)_d^2} \\[4mm]
\beta_{22} = \dfrac{\sum\limits_{d=1}^{29} (x'_{2d}) y_d}{\sum\limits_{d=1}^{29} (x_{2d})^2}, \beta_{23} = \dfrac{\sum\limits_{d=1}^{29} (x_2 x_3)_d y_d}{\sum\limits_{d=1}^{29} (x_2 x_3)_d^2}, \beta_{33} = \dfrac{\sum\limits_{d=1}^{29} (x_{3d}) y_d}{\sum\limits_{d=1}^{29} (x_{3d})^2}
\end{cases} \tag{5}$$

**Table 6. Model verification results.**

| Serial number | $X_1$ (rpm) | $X_2$ (mm) | $X_3$ (mm) | $X_4$ (mm) | Experimental (μm) | Estimate (μm) |
|---|---|---|---|---|---|---|
| A | 2500 | 5 | 0.1 | 0.04 | 0.175 | 0.169 |
| B | 1500 | 8 | 0.1 | 0.03 | 0.293 | 0.302 |
| C | 500 | 2 | 0.1 | 0.01 | 0.341 | 0.334 |

According to Eqs 1–5 and the data in the table, the RSM prediction model of the milling surface roughness of aerospace titanium alloy TI64 specimen can be obtained as follows:

$$Ra = 0.38 + (-0.01)X_1 + 0.075X_2$$
$$+0.15X_3 + (-0.14)X_4 + (-0.077)X_1X_2$$
$$+0.00575X_1X_3 + (-0.00025)X_1X_4 \tag{6}$$
$$+0.026X_2X_3 + 0.00025X_2X_4 + X_3X_4$$
$$-0.014X_1^2 - 0.016X_2^2 + 0.031X_3^2 + 0.068X_4^2$$

Where: $R_a$ is the milling surface roughness of Ti64 specimen, μm; $X_1$, $X_2$, $X_3$ and $X_4$ are respectively the spindle speed, cutting width, cutting depth and feed per tooth.

In order to verify the accuracy of the roughness prediction model, milling experiments were carried out on the experimental platform in the second part by selecting three groups of different milling parameters in Table 6.

Fig 2 shows the comparison between the predicted values of the model and the actual values. Each scatter point is approximately distributed near the same line, which proves that the established regression model has a high significance level and accurate prediction results.

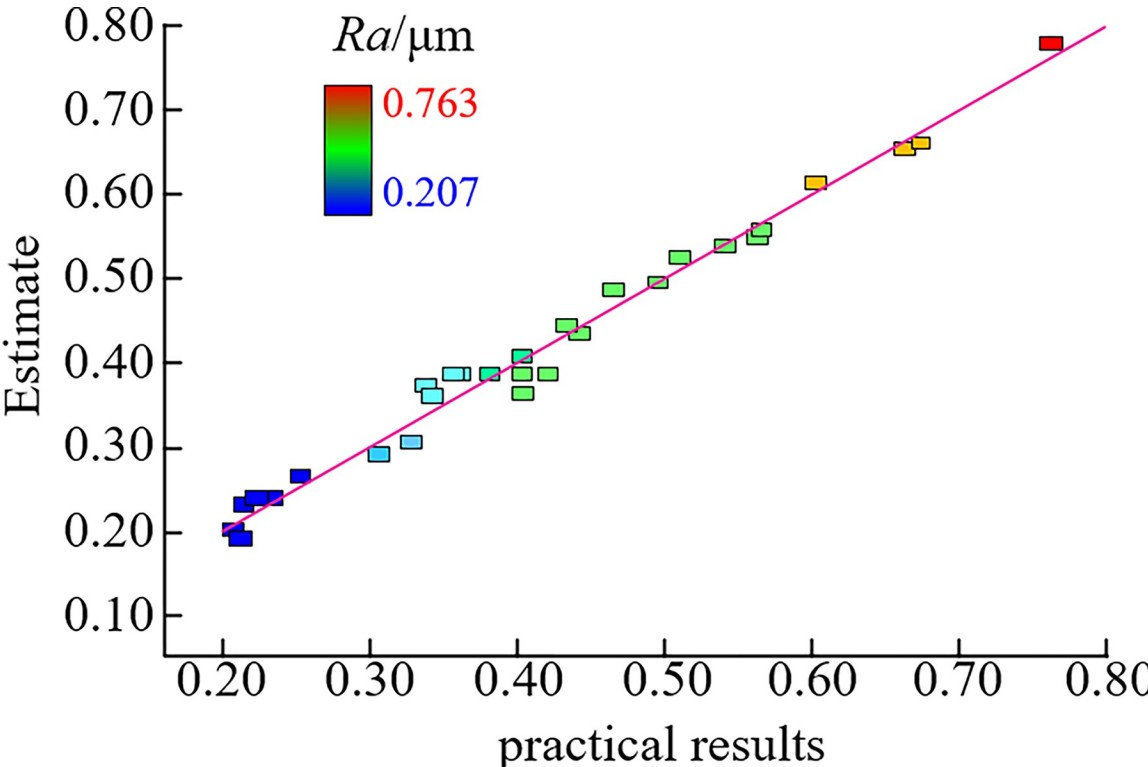

**Fig 2. Comparison of predicted values with actual values.**

**Table 7. Model verification results.**

| Serial number | $X_1$ (rpm) | $X_2$ (mm) | $X_3$ (mm) | $X_4$ (mm) | Experimental (μm) | Estimate (μm) |
|:---:|:---:|:---:|:---:|:---:|:---:|:---:|
| A | 2500 | 5 | 0.1 | 0.04 | 0.175 | 0.169 |
| B | 1500 | 8 | 0.1 | 0.03 | 0.293 | 0.302 |
| C | 500 | 2 | 0.1 | 0.01 | 0.341 | 0.334 |

As shown in Table 7, comparing the milling surface roughness experimental value and the model predicted value, we can see that the relative error of the experimental value and the model predicted value is less than 3.5%, the maximum error is only 0.9%, and the average error is 0.7%. The verification results show that the regression model can accurately predict the milling surface roughness of the specimen.

# 3 Snake algorithm

The SO algorithm implements global optimization of the optimization algorithm by simulating the fighting mode and mating mode of snakes, in which each male fights to get the best female and each female tries to choose the best male. In mating mode, mating takes place between males and females with a availability coefficient relative to the amount of food available. If the mating process occurs within the search phase, the female snake is likely to lay eggs, hatching new offspring.

## 3.1 Initialization

SO first generates an evenly distributed random population, and assumes that the proportion of females and males in the snake population is 50%, and the population is divided into male and female populations.

## 3.2 The algorithm is divided into stages

SO algorithm optimization process is divided into exploration stage and development stage. The exploration stage simulates the behavior pattern of snakes in the absence of food, and the development stage simulates the behavior pattern of snakes in the presence of food. The behavior pattern of snakes is controlled by the total amount of material $Q$ and the temperature $Temp$ [23].

1. The exploration stage
   If $Q < Threshold_Q(0.25)$, snakes find food by selecting any random location and updating their location. The equation for location update is as follows:

$$X_{i,m}(t+1) = X_{rand,m}(t) \pm X$$
$$X = c_2 \times A_m \times ((X_{\max} - X_{\min}) \times rand + X_{\min}) \tag{7}$$

   Where $X_{i,m}$ represents the position of the $i$ male, $X_{rand,m}$ represents the position of the random male, $A_m$ represents the predatory ability of the individual male snake, $rand \in (0, 1)$, $C_2$ = 0.05.

2. Development stage

When $Q < Threshold_Q$, if Temp $> Threshold_{Temp}$ (0.6), the snake will only move towards the food, and the movement equation for the individual snake is as follows:

$$X_{i,j}(t+1) = X_{food} \pm X$$
$$X = c_3 \times Temp \times rand \times (X_{food} - X_{i,j}(t)) \tag{8}$$

Where $X_{i,j}$ is the position of individual $i$(male or female), $X_{food}$ is the position of the best individual, and $C_1$ is constant and equal to 2.

If $Temp <$ Threshold$_{Temp}$ (0.6), the snake will be in battle mode or mating mode, and the equation for the movement of an individual snake in battle mode is as follows:

$$X_{i,m}(t+1) = X_{i,m}(t) + X$$
$$X = c_3 \times FM \times rand \times (Q \times X_{best,f} - X_{i,m}(t)) \tag{9}$$

Where $X_{i,m}$ is the position of male individual $i$, $X_{food}$ is the best position in the female group, and $FM$ is the male combat effectiveness.

When $Temp <$ Threshold$_{Temp}$ (0.6), the equation for the movement of an individual snake in mating mode is as follows:

$$X_{i,m}(t+1) = X_{i,m}(t) + X$$
$$X = c_3 \times M_m \times rand \times (Q \times X_{i,f}(t) - X_{i,m}(t)) \tag{10}$$

Where $X_{i,f}$ is the position of individual $i$ in the female group, $X_{i,f}$ is the position of individual $i$ in the male group, and $M_m$ is the mating ability of the male.

## 4 Improve the snake algorithm

### 4.1 Improvements to initialize the population

In order to increase the diversity of the initial population of the algorithm and reduce the overall and local density of the initial population as much as possible, the initial solution should be uniformly generated so that the algorithm can evenly explore the entire search space [24]. Orthogonal array can provide a uniform distribution of position combinations, using this property of the orthogonal table to construct the initial solution, the orthogonal matrix is expressed as $L_Q (N^M)$, the size of the matrix is $Q \times M$, which contains $M$ factors, each factor is divided into $N$ levels. In this paper, the orthogonal matrix $L_Q (N^M)$ is constructed by using the method proposed in the above paper, where $N$ is an odd integer, $Q = N^I$, and the way $I$ is selected satisfies the equation:

$$N = \frac{Q^I - 1}{Q - 1} \tag{11}$$

The orthogonal table $A$ is constructed according to the following steps
Step 1: Calculate the base elements of $A$:

$$\alpha_{ij} = \lfloor * \rfloor \frac{i-1}{N^{I-k}}, k = 1 \text{ to } I \text{ and } i = 1 \text{ to } Q \tag{12}$$

$$j = \frac{N^{k-1} - 1}{N - 1} + 1 \tag{13}$$

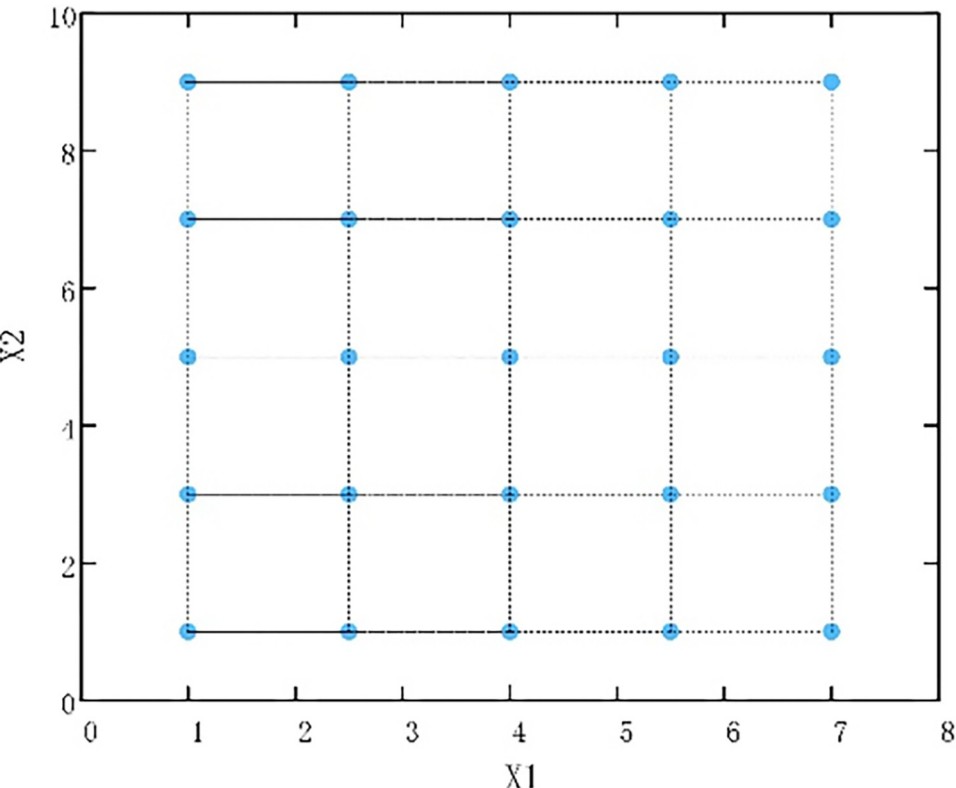

**Fig 3. Example of orthogonal initialization.**

Step 2: Calculate the non-basic elements of $A$:

$$\boldsymbol{\alpha}_{j+(s-1)(N-1)+t} = \mathrm{rem}(\boldsymbol{\alpha}_s t + \boldsymbol{\alpha}_j, N),$$
$$s = 1 \mathrm{\,to}(j-1) \mathrm{\;and\;} t = 1 \mathrm{\;to\;} (N-1) \tag{14}$$

$$\boldsymbol{\alpha}_j = \left[\boldsymbol{\alpha}_{1j}, \boldsymbol{\alpha}_{2j}, \boldsymbol{\alpha}_{3j}, \ldots, \boldsymbol{\alpha}_{Qj}\right]^T \tag{15}$$

Step 3: Remove the last ($M$-$D$) column of $A$ so that the matrix is controlled in column $D$.

Step 4: Randomly delete rows ($N$-$Q$) of $A$ so that the matrix control is in row $Q$.

Calculate to get the matrix and generate the initial solution using the following equation:

$$x_{ij} = \alpha_{ij}\left(\frac{x_U - x_L}{max(A) - min(A)}\right) + x_L,$$
$$i = 1, 2, \ldots, Q \; and \; j = 1, 2, \ldots, D \tag{16}$$

When the dimension of the solution space is 2, $lb$ = [1.0, 1.0], $ub$ = [7.0, 9.0], and the factor is 5. The 25 individuals obtained by orthogonal initialization are shown in Fig 3.

## 4.2 Analysis and improvement of food quantity and temperature mechanism

According to Chapter 3, SO algorithm is divided into exploration stage and development stage. Fig 4A shows the two-dimensional space movement trajectory of the snake individual in an iteration in the exploration stage. Fig 4B–4D represent the two-dimensional space motion

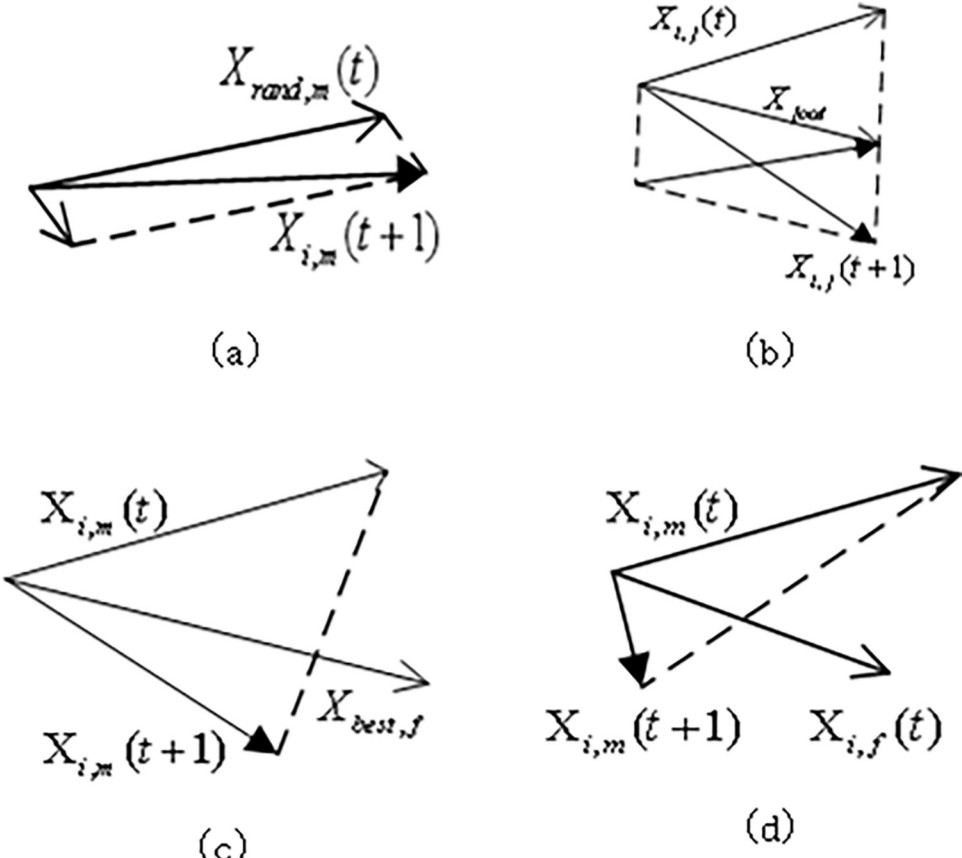

**Fig 4. Snake optimizer individual movement trajectory map.** (a) Eq (1) individual motion trajectory. (b) Eq (2) individual motion trajectory. (c) Eq (3) individual motion trajectory. (d) Eq (4) individual motion trajectory.

trajectories corresponding to Eqs (2)–(4) of individual snake in the development stage, respectively. It can be seen from Fig 4A that the position of the individual snake after updating is basically the same as that before updating, indicating that when SO algorithm is in the exploration stage, the individual snake moves in its own small range. It can be seen from Fig 2B that in order to find food, the individual snake will move closer to the food, and the food represents the position of the best individual. As can be seen from Fig 4C, in order to obtain sufficient food when entering the fighting mode, individual snakes will seek the strongest opposite sex. As can be seen from Fig 4D, when entering the mating mode, individual snakes will change their movement rules and get closer to the opposite sex. It can be seen that in terms of the underlying structure of the model of the snake algorithm, the exploration stage is used for global optimization, and the development stage is used for local optimization and jumping out of local optimal solutions. The design architecture of the algorithm is relatively complete, but the phase transformation cannot be carried out because the food quantity and temperature do not exceed the threshold value, making the interaction between the stages too rigid (Fig 4).

Food quantity $Q$ is a key parameter to balance the proportion of SO in the exploration stage and the development stage. The description of $Q$ in SO is from the minimum value to the maximum value, and SO switches from the exploration stage to the development stage when it drops to a certain threshold. This way of switching may result in too long a time in the exploration phase, leaving insufficient time for the development phase, and the temperature threshold of the control phase switching may be different for different problems. In order to make the

design of $Q$ more flexible and reduce the proportion of SO in the exploration stage, the fitness equation is used to replace the simple assignment of food quantity in the original algorithm to ensure the dynamic change of food quantity, so as to achieve the effect of balancing the proportion in the algorithm stage. The expression is shown as follows.

$$Q = |gbest_t - gbest_{t-1}| > \frac{|gbest_t - gbest_0|}{t} \tag{17}$$

The equation of temperature *Temp* is to switch the position update method of snake individuals according to the current number of iterations, and restart the population to a certain extent to ensure that the algorithm does not converge prematurely. According to the following equation, the change strategy of *Temp* is optimized, and the conditions of the algorithm to restart the population are optimized.

$$Temp = \frac{\sum_{i=1}^{n}(\text{pbest}_i - p\bar{best})}{n_0} < k \tag{18}$$

Where *Temp* represents the mean variance of each individual in the population in each dimension, $k$ is the constant threshold, $n_0$ is the number of individuals in the population. The Eq (18) determines the degree of aggregation of individuals, and calculates the result of each individual to determine whether the variance is less than $k$. If the conditions are met, it means that most particle search results are similar, indicating that the iterative results are convergent, SO you can directly switch to the development stage to speed up the convergence, and then switch to the search stage when the above equation is not established.

## 4.3 Joint reverse selection

Combined with the advantages of selective dominant opposition and dynamic opposition [24], joint reverse selection enhances the exploration and development ability of the algorithm, enhances the global distribution of snake individuals, and improves the optimization ability of the algorithm.

**4.3.1 Selective dominant opposition.** SLO [25] (Selective Leading Opposition, SLO), developed from Selective Opposition, calculates the difference distance in each dimension between the current solution and the optimal solution and compares it to the threshold, Dimensions greater than the threshold are the distance dimensions ($d_f$), and those less than the threshold are the proximity dimensions ($d_c$). At the same time, it is also necessary to calculate the Spearman phase relation value between the current solution and the optimal solution, and execute the SLO strategy for the position where the correlation is less than 0. The SLO policy is calculated by the following equation.

$$\bar{X}_{i,d_c} = lb + ub - X_{i,d_c}$$
$$if \; src < 0 \; and \; size(d_c) > size(d_f) \tag{19}$$

$$src = 1 - \frac{6 \times \sum_{j=1}^{\dim}(dd_{i,j})^2}{n \times (n^2 - 1)} \tag{20}$$

Where $X_{i,d_c}$ is the approach dimension of the ith solution and $\bar{X}_{i,d_c}$ is the opposite solution of the approach dimension. *src* is the Spearman rank phase relation value. *Size ($d_c$)* is the number of proximity dimensions of the currentc solution. *Size ($d_f$)* is the number of distant

dimensions of the currentf solution. Calculate the difference distance of each dimension between the current solution and the optimal solution by Eq (21). The dimension where the difference is less than the threshold is $d_c$, and the opposite dimensionc is $d_f$, and the threshold is calculated by Eq (22).

$$dd_{i,j}{}^{t} = |X_{best,j}{}^{t} - X_{i,j}{}^{t}| \tag{21}$$

$$threshold = 2 - \frac{2t}{T} \tag{22}$$

**4.3.2 Dynamic opposition.**   Dynamic opposite (DO) combines the ideas of quasi-opposition [26] and quasireflection [27]. It has the advantage that it can search the space dynamically, move asymmetrically in the search space, and help the algorithm escape from the local optimal solution, the calculation equation is as follows.

$$\bar{X}_i = lb + ub - X_i \tag{23}$$

$$X_r = rand \times \bar{X}_i \tag{24}$$

$$\bar{X}_{do} = X_i + rand \times (X_r - X_i)$$
$$if \ rand < jr \tag{25}$$

Where, $\bar{X}_i$ is the inverse solution of $X_i$ $X_i$ is the ith solution. $X_r$ is the random opposite solution. $X_r$ is the opposite solution of dynamic $\bar{X}_i$. $jr$ is the jump rate, which is the probability of performing a dynamic opposition. Ref. [28] states that a value of 0.25 works best.

## 4.4 Improve the logical flow of the algorithm

Step 1: initialization, maximum number of iterations $T$, population individual $N$, objective equation dimension $D$, boundary $lb$, $ub$.

Step 2: Use the Eq (16) for orthogonal population initialization to generate $N$ snake individuals $X = [X_{i1}, X_{i2}, ..., X_{iD}]$. Where $X_{iD}$ represents the position of the $i$ snake in $D$ dimension; Male and female populations are divided, and individual fitness and optimal fitness of male and female populations are calculated.

Step 3: Update the positions of all individuals through the selective dominant opposition strategy. If Eq (17) is not true, enter the exploration mode, otherwise enter the development mode.

Step 4: In the development mode, $Temp > k$, update the position of snake individual according to Eq (8). If $Temp < k$, the individual snake will fight or mate.

Step 5: When the individual snake is in battle mode, it updates its position by Eq (9), and when it is in mating mode, it updates its position by Eq (16).

Step 6: When individual snakes mate and produce offspring, all individual snakes restart through Eq (25).

Step 7: Update individual positions, calculate the current fitness values of all individuals, update the current male and female population and the global optimal fitness. If the maximum number of iterations T is reached, go to step 8. Otherwise, go to step 3.

Step 8: Output the optimal fitness and position $X_{best}$.

**Table 8. Parameters of each algorithm.**

| Algorithmic name | Algorithmic parameter |
|---|---|
| MSSO | - |
| SO | $c_1 = 0.5$, $c_2 = 0.05$, $c_3 = 2$ |
| WOA | $\alpha = 2{\sim}0$, $b = 2$ |
| SCA | - |
| GWO | $\alpha = 2{\sim}0$ |
| LDWPSO | $w_{min} = 0.4$, $w_{max} = 0.9$, $c1 = c2 = 2$ |
| MPSO | $w_{min} = 0.4$, $w_{max} = 0.9$, $c1 = c2 = 2$ |

## 4.5 Experiment and experimental analysis

**4.5.1 Benchmark test equation and experimental environment.** In order to verify the correct line selection of MSSO algorithm optimization strategy, 10 classic single-peak and multi-peak benchmark test equations were selected to evaluate the performance of MSSO. The experiment was carried out in Windows10, Inter Core i5-12490F environment, and the programming language is MATLAB2020b. The parameters of the test algorithm are shown in Table 8 and the test results are shown in Table 9.

**4.5.2 Comparison of MSSO algorithm with other intelligent algorithms.** In order to test the performance of MSSO, it is compared with snake algorithm SO [11], whale algorithm WOA [29], sine-cosine algorithm SCA [30], grey wolf algorithm GWO [31], linear decreasing inertia weight particle swarm algorithm LDWPSO [32], improved particle swarm algorithm MPSO [33], and the iteration number T is set to 500 times. The population size N is 50. The

**Table 9. Baseline test equations.**

| equation names | Expression | Dimension | Scope of search | Minimum value |
|---|---|---|---|---|
| F1 | $f(x) = \sum_{i=1}^{n} x_i^2$ | 10 | $[-100,100]$ | 0 |
| F2 | $f(x) = \sum_{i=0}^{n} \lvert x_i \rvert + \prod_{i=0}^{n} \lvert x_i \rvert$ | 10 | $[-10,10]$ | 0 |
| F3 | $f(x) = \sum_{i=1}^{d} \left( \sum_{j=1}^{i} x_j \right)^2$ | 10 | $[-100,100]$ | 0 |
| F4 | $f(x) = \max_i \{ \lvert x_i \rvert, 1 in \}$ | 10 | $[-100,100]$ | 0 |
| F5 | $f(x) = \sum_{i=1}^{n} [x_i^2 - 10\cos(2\pi x_i) + 10]$ | 10 | $[-5.12,5.12]$ | 0 |
| F6 | $f(x) = -20 \exp\left( -0.2 \sqrt{\frac{1}{n}\sum_{i=1}^{n} x_i^2} \right) - \exp\left( \frac{1}{n}\sum_{i=1}^{n} \cos(2\pi x_i) \right) + 20 + e$ | 10 | $[-32,32]$ | 0 |
| F7 | $f(x) = 1 + \frac{1}{4000}\sum_{i=1}^{n} x_i^2 - \prod_{i=1}^{n} \cos\left( \frac{x_i}{\sqrt{i}} \right)$ | 10 | $[-600,600]$ | 0 |
| F8 | $f(x) = -\sum_{i=1}^{4} c_i \exp\left( -\sum_{i=1}^{6} a_{ij}(x_j - p_{ij})^2 \right)$ | 6 | $[-0,1]$ | $-0.32$ |
| F9 | $f(x) = -\sum_{i=1}^{5} \left[ (X - a_i)(X - a_i)^T + c_i \right]^{-1}$ | 4 | $[-0,1]$ | $-10.153$ |
| F10 | $f(x) = \sum_{i=1}^{11} \left[ a_i - \frac{x_1(b_i^2 + b_i x_2)}{b_i^2 + b_i x_3 + x_4} \right]^2$ | 10 | $[-5,5]$ | 0.0003 |

**Table 10. Optimum fitness of the test equationTest sequence number.**

| Measured values | Comparison algorithm | | | | | | |
|---|---|---|---|---|---|---|---|
| | MSSO | GWO | SCA | WOA | MPSO | SO | LDWPSO |
| F1 | | | | | | | |
| Mean | 0.0000E+00 | 2.1575E-70 | 3.7184E-16 | 6.8548E-95 | 7.3647E-48 | 1.6452E-120 | 4.1984E+01 |
| STD | 0.0000E+00 | 4.2361E-68 | 6.1236E-15 | 3.1868E-88 | 8.2780E-37 | 1.5869E-118 | 2.9344E+00 |
| F2 | | | | | | | |
| Mean | 1.2972E-121 | 7.9395E-40 | 1.5448E-11 | 1.6606E-57 | 1.2707E-23 | 2.1632E-62 | 5.2436E+01 |
| STD | 8.0255E-120 | 1.6421E-40 | 2.3111E-11 | 1.7047E-58 | 2.3172E-23 | 8.1841E-61 | 2.1368E+01 |
| F3 | | | | | | | |
| Mean | 0.0000E+00 | 6.8630E-33 | 2.3989E-10 | 1.8321E-03 | 3.8540E-19 | 1.8007E-88 | 4.8510E+01 |
| STD | 0.0000E+00 | 8.1354E-32 | 2.1500E-03 | 4.2985E+01 | 4.9907E+01 | 1.7495E-84 | 7.5946E+00 |
| F4 | | | | | | | |
| Mean | 1.4753E-111 | 3.1203E-22 | 7.0094E-06 | 2.2369E+00 | 2.1416E-15 | 2.7728E-54 | 5.9024E+01 |
| STD | 6.1430E-110 | 2.6421E-20 | 5.1399E-05 | 2.5493E-01 | 5.8635E-15 | 8.8088E-53 | 6.3484E-02 |
| F5 | | | | | | | |
| Mean | 0.0000E+00 | 0.0000E+00 | 2.2558E-02 | 0.0000E+00 | 7.4774E+00 | 0.0000E+00 | 5.0844E+01 |
| STD | 0.0000E+00 | 0.0000E+00 | 1.3210E+00 | 0.0000E+00 | 9.5764E+00 | 0.0000E+00 | 4.4586E+00 |
| F6 | | | | | | | |
| Mean | 8.8818E-16 | 8.8818E-16 | 8.8818E-16 | 4.4409E-15 | 4.4409E-15 | 8.8818E-16 | 4.5825E+01 |
| STD | 0.0000E+00 | 0.0000E+00 | 0.0000E+00 | 1.8346E-15 | 2.1965E-05 | 0.0000E+00 | 2.3828E+01 |
| F7 | | | | | | | |
| Mean | 0.0000E+00 | 0.0000E+00 | 1.5788E-02 | 0.0000E+00 | 1.0437E-02 | 0.0000E+00 | 4.2722E+02 |
| STD | 0.0000E+00 | 0.0000E+00 | 1.3210E+00 | 0.0000E+00 | 9.5764E+00 | 0.0000E+00 | 4.4586E+00 |
| F8 | | | | | | | |
| Mean | -3.3220E+00 | -3.2027E+00 | -3.0144E+00 | -3.1907E+00 | -3.3179E+00 | -3.2031E+00 | 5.2423E+01 |
| STD | 0.0000E+00 | 0.0000E+00 | 0.0000E+00 | 0.0000E+00 | 0.0000E+00 | 0.0000E+00 | 8.1528E+00 |
| F9 | | | | | | | |
| Mean | -1.0153E+01 | -1.0152E+01 | -8.8093E-01 | -1.0153E+01 | -9.6106E+00 | -1.0151E+01 | 1.4509E+01 |
| STD | 0.0000E+00 | 0.0000E+00 | -7.4232E-01 | 0.0000E+00 | -2.7532E+00 | 0.0000E+00 | 0.0000E+00 |
| F10 | | | | | | | |
| Mean | 3.0746E-04 | 2.0364E-02 | 6.2078E-04 | 6.7696E-04 | 1.4948E-03 | 4.2511E-04 | 5.3792E-04 |
| STD | 2.1683E-03 | 1.8409E-02 | 5.3481E-04 | 2.8943E-04 | 9.7615E-02 | 2.7591E-04 | 3.0921E-04 |
| RANK | 1 | 3 | 5 | 4 | 6 | 2 | 7 |
| Avg RANK | 1 | 3 | 5 | 3.6 | 6.1 | 1.9 | 6.4 |

mean and standard deviation (STD) obtained by 30 independent execution of the above selected algorithm were compared. The parameter configuration of the selected algorithm is shown in Table 10, the operation results and comprehensive ranking are shown in Table 10, and the best fitness curve of each algorithm is shown in Fig 5.

**4.5.3 Effectiveness analysis of the improvement strategy.** In order to further verify the effectiveness of the three improved strategies, the test equation in Table 8 is used to test the standard SO, only SO (OSO) initialized by orthogonal matrix, SO (DQSO) with dynamic adaptive parameters, and SO (JOSO) and MSSO with joint reverse selection under the conditions of 10, 50 and 100 dimensions respectively [34]. Considering the length factor, 3 unimodal equations and 2 multi-modal equations are selected from Table 8 for testing. The parameter Settings of each algorithm are consistent with SO, and the results of each algorithm after running independently for 30 times are shown in Table 11.

As can be seen from Table 11, from the perspective of optimal and average values, MSSO can find the theoretical optimal value when solving F1, F5, and F6 in the three dimensions.

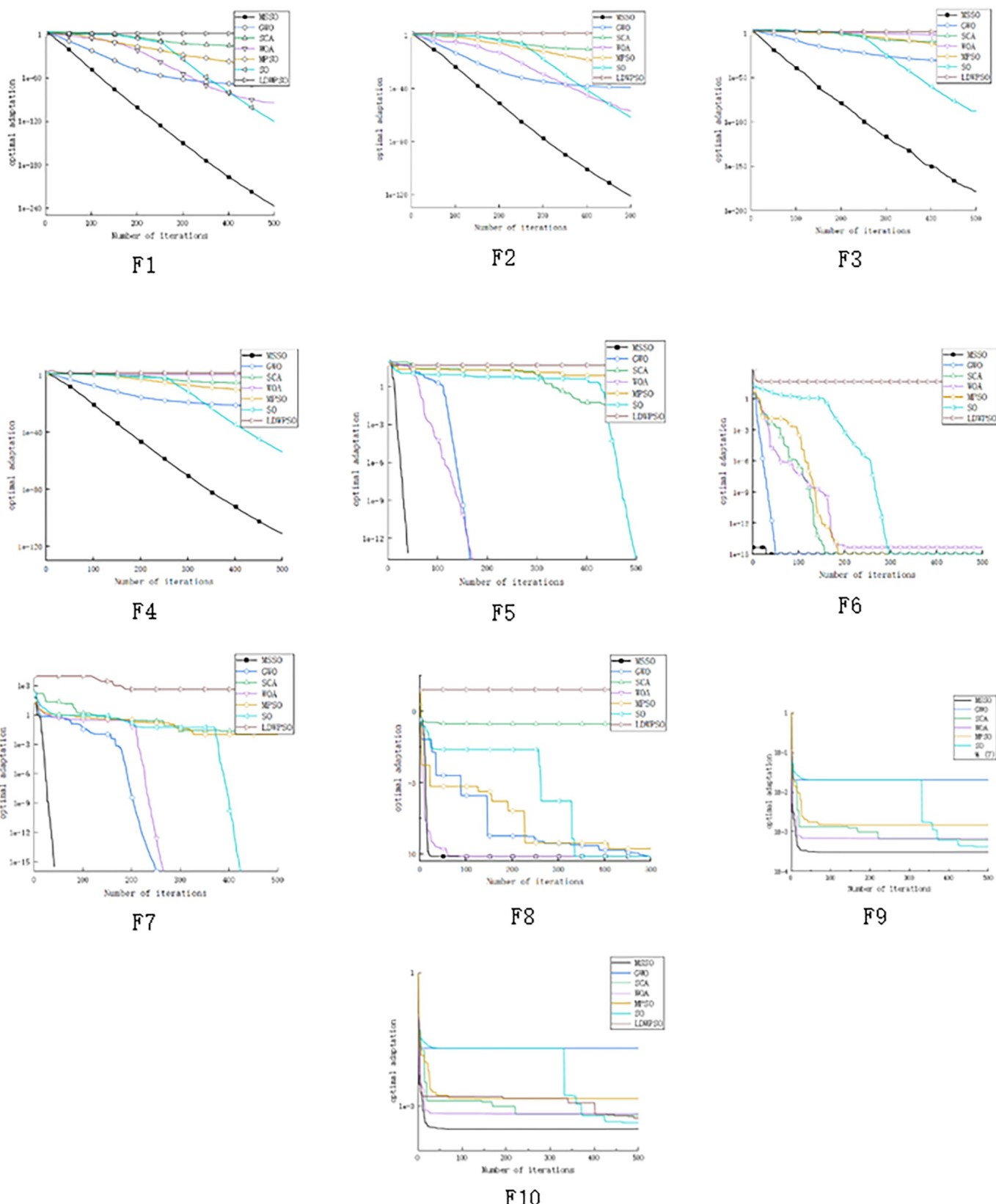

**Fig 5. Convergence curve of benchmark equation F1-F10.** (a) F1. (b) F2. (c) F3. (d) F4. (e) F5. (f) F6. (g) F7. (h) F8. (i) F9. (k) F10.

**Table 11. Optimization results of different improvement strategies in different dimensions.**

| Statistics | Algorithms | D = 10 | | | D = 50 | | | D = 100 | | |
|---|---|---|---|---|---|---|---|---|---|---|
| | | Best | Mean | STD | Best | Mean | STD | Best | Mean | STD |
| F1 | SO | 3.12E-119 | 1.55E-118 | 5.36E-117 | 2.77E-86 | 1.65E-85 | 8.52E-84 | 1.63E-84 | 2.10E-83 | 7.37E-82 |
| | OSO | 7.12E-121 | 9.05E-120 | 7.34E-119 | 2.54E-90 | 1.27E-88 | 6.23E-86 | 7.31E-90 | 5.93E-90 | 1.63E-89 |
| | DQSO | 5.43E-129 | 1.97E-128 | 1.55E-127 | 7.34E-92 | 4.12E-92 | 8.34E-91 | 6.41E-94 | 1.98E-93 | 7.17E-92 |
| | JOSO | 7.44E-123 | 1.22E-122 | 7.33E-121 | 1.64E-92 | 2.57E-91 | 9.11E-90 | 7.85E-86 | 2.10E-86 | 8.29E-85 |
| | MSSO | 0.00E+00 | **0.00E+00** | 0.00E+00 | 0.00E+00 | **0.00E+00** | 0.00E+00 | 0.00E+00 | **0.00E+00** | 0.00E+00 |
| F4 | SO | 8.78E-54 | 4.13E-54 | 1.84E-53 | 8.66E-41 | 6.32E-41 | 1.76E-40 | 3.87E-40 | 5.49E-39 | 7.24E-38 |
| | OSO | 8.23E-58 | 2.76E-58 | 7.11E-57 | 7.10E-59 | 1.15E-59 | 7.63E-58 | 6.11E-50 | 5.16E-49 | 8.47E-48 |
| | DQSO | 6.84E-105 | 1.32E-105 | 7.03E-104 | 8.49E-64 | 6.47E-64 | 1.62E-63 | 7.47E-83 | 1.43E-83 | 8.64E-82 |
| | JOSO | 7.61E-63 | 2.41E-62 | 1.85E-61 | 7.81E-52 | 7.02E-51 | 9.71E-50 | 2.10E-58 | 7.56E-57 | 7.18E-56 |
| | MSSO | 7.22E-113 | **1.21E-113** | 3.49E-112 | 3.98E-98 | **6.26E-97** | 2.90E-96 | 5.90E-95 | **6.63E-95** | 9.71E-94 |
| F5 | SO | 2.55E-08 | 1.53E-08 | 7.51E-07 | 3.71E+00 | 1.28E+00 | 6.27E00 | 1.34E+00 | 3.29E+00 | 7.90E+00 |
| | OSO | 0.00E+00 | 0.00E+00 | 0.00E+00 | 0.00E+00 | 0.00E+00 | 0.00E+00 | 5.96E-01 | 1.01E+00 | 6.83E+00 |
| | DQSO | 0.00E+00 | 0.00E+00 | 0.00E+00 | 0.00E+00 | 0.00E+00 | 0.00E+00 | 0.00E+00 | 0.00E+00 | 0.00E+00 |
| | JOSO | 3.83E-10 | 1.67E-10 | 7.49E-9 | 2.85E-02 | 6.02E-01 | 9.47E-00 | 6.74E-01 | 2.70E+00 | 6.38E+00 |
| | MSSO | 0.00E+00 | **0.00E+00** | 0.00E+00 | 0.00E+00 | **0.00E+00** | 0.00E+00 | 0.00E+00 | **0.00E+00** | 0.00E+00 |
| F6 | SO | 8.88E-16 | 8.88E-16 | 0.00E+00 | 4.44E-16 | 1.23E-01 | 9.40E-01 | 4.44E-15 | 1.30E-14 | 9.21E-13 |
| | OSO | 8.88E-16 | 8.88E-16 | 0.00E+00 | 4.44E-16 | 4.44E-16 | 0.00E+00 | 4.44E-15 | 4.14E-14 | 8.83E-13 |
| | DQSO | 8.88E-16 | 8.88E-16 | 0.00E+00 | 4.44E-16 | 1.36E-15 | 1.28E-14 | 4.44E-15 | 2.62E-14 | 2.18E-13 |
| | JOSO | 8.88E-16 | 8.88E-16 | 0.00E+00 | 4.44E-16 | 1.48E-15 | 4.21E-14 | 4.44E-15 | 7.91E-14 | 6.23E-13 |
| | MSSO | 8.88E-16 | **8.88E-16** | 0.00E+00 | 8.88E-16 | **8.88E-16** | 0.00E+00 | 8.88E-16 | **6.93E-15** | 4.98E-14 |
| F10 | SO | 3.41E-04 | 3.78E-04 | 1.39E-4 | 6.91E-04 | 1.13E-03 | 8.28E-02 | 1.55E-03 | 1.72E-03 | 7.81E-03 |
| | OSO | 3.23E-04 | 3.42E-04 | 2.17E-3 | 4.21E-04 | 4.51E-04 | 2.76E-03 | 4.11E-04 | 4.59E-04 | 5.71E-03 |
| | DQSO | 3.39E-04 | 3.54E-04 | 2.61E-4 | 4.11E-04 | 4.57E-04 | 6.98E-03 | 4.29E-04 | 4.71E-04 | 2.72E-04 |
| | JOSO | 3.11E-04 | 3.19E-04 | 1.75E-3 | 3.68E-04 | 4.18E-04 | 1.98E-04 | 3.91E-04 | 4.30E-04 | 9.12E-05 |
| | MSSO | 3.01E-04 | **3.01E-04** | 0.00E+00 | 3.01E-04 | **3.06E-04** | 1.73E-05 | 3.01E-04 | **3.15E-04** | 8.11E-05 |

Although the theoretical optimal value is not found, their convergence accuracy is more than ten orders of magnitude better than SO. For the function F4, the accuracy of MSSO solution results is optimal in 10 dimensions, and the optimization accuracy of DQSO and MSSO in 50 and 100 dimensions is of the same order of magnitude and superior to the comparison model, indicating that the convergence accuracy of the algorithm is improved by introducing dynamic parameters $Q$ and $Temp$. For the function F7, the optimization accuracy of JOSO and MSSO is relatively high, indicating that the introduction of joint reverse selection can increase the optimization ability of the algorithm, and improve the anti-stagnation and global exploration ability of the algorithm. Secondly, from the perspective of standard deviation, the standard deviation of MSSO, OSO, DQSO, and JOSO are all smaller than SO when solving the 5 test equation, and the standard deviation of MSSO when solving F1, F5, and F6 is 0, indicating that each improvement strategy has different degrees of improvement on the stability of the algorithm. However, under a single improvement strategy, the optimization effect of the SO algorithm on different test equations is not stable enough, and the improvement accuracy of some test equations is limited, indicating that the introduction of only a single improvement strategy cannot meet the requirements of high optimization accuracy and strong robustness of the algorithm. However, MSSO with three improved strategies can improve the optimization accuracy and robustness significantly under different test equations and different dimensions.

**4.5.4 Analysis of MSSO population diversity.** To verify the influence of the improved strategy on the population diversity and the convergence of the improved algorithm during

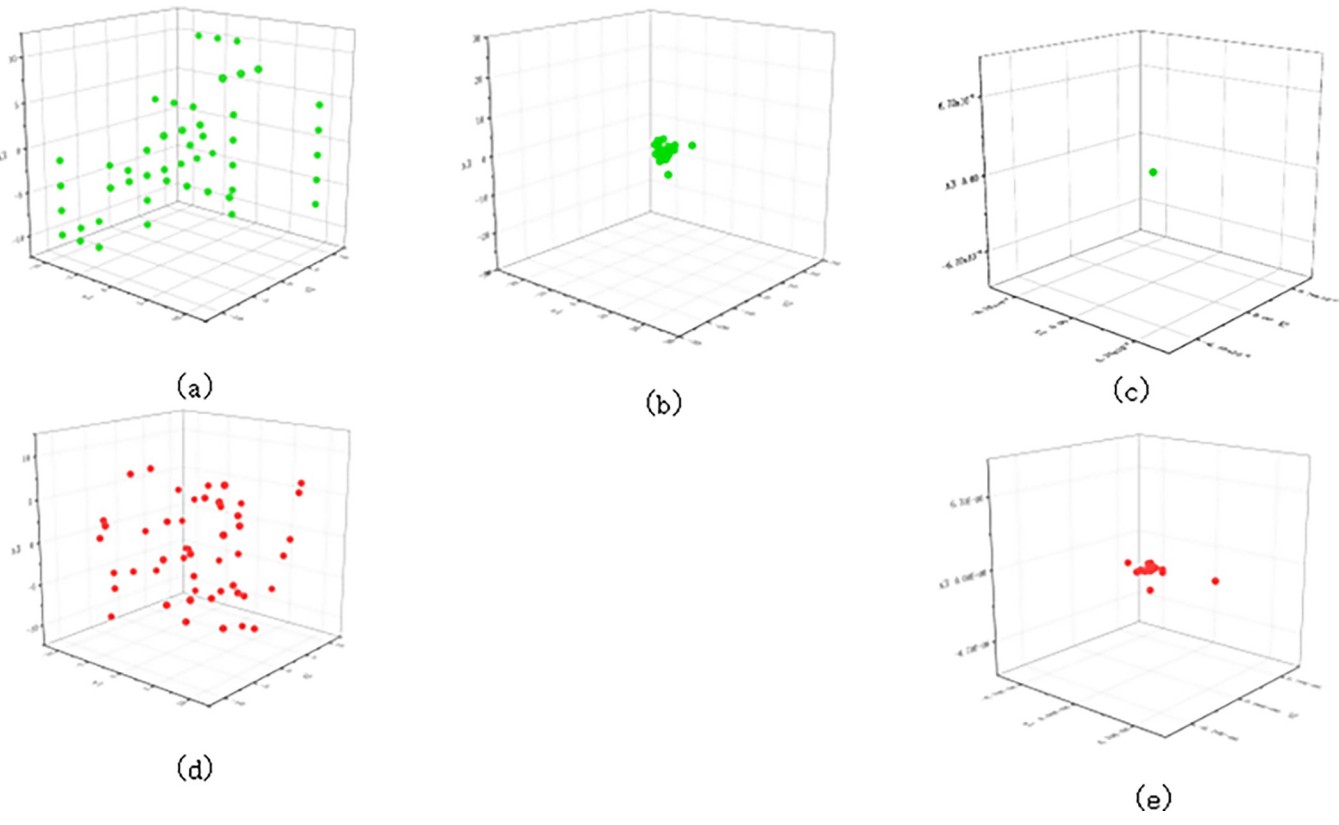

**Fig 6. Population distribution map of MSSO and SO.**

iteration, the function F1 test was used to conduct optimization experiments, and the problem dimension $D = 3$ [35]. The population distribution after MSSO initialization is shown in Fig 6A. The distribution map of snake individuals after 10 and 50 iterations of MSSO is drawn and compared with the position of population individuals after SO initialization and 10 and 50 iterations.

As can be seen from Fig 6A and 6D, MSSO uses the initialization strategy of the orthogonal matrix to avoid the problem of low population diversity and uneven distribution caused by the SO algorithm adopting random distribution for the initial population. From Fig 6A–6C, it can be seen that snake individuals gather to the optimal value at a faster speed with the iteration of MSSO, and the distribution of population individuals can still maintain a good uniformity after iteration. Compared with Fig 6C and 6E, it can be seen that the population individuals have been distributed in the optimal solution position after 50 iterations of MSSO. However, the SO population has not completed convergence, indicating that the improvement strategy effectively improves the diversity of the initial population and the convergence speed of the algorithm.

## 5 Experimental verification

The improved algorithm was used to optimize the prediction model of milling surface roughness of titanium alloy, and minRa = 0.105, $X_1$ = 3018rpm, $X_2$ = 2mm, $X_4$ = 0.3mm, $X$ = 0.02mm were obtained.

To verify the feasibility and effectiveness of the optimization of milling parameters by the MSSO algorithm, comparative experiments were carried out between the parameters used

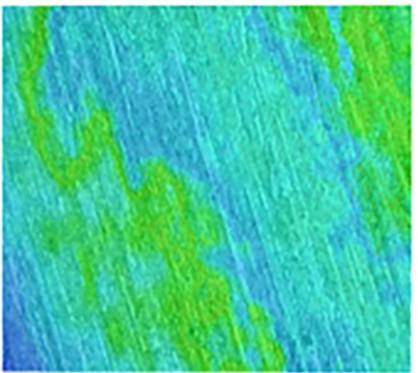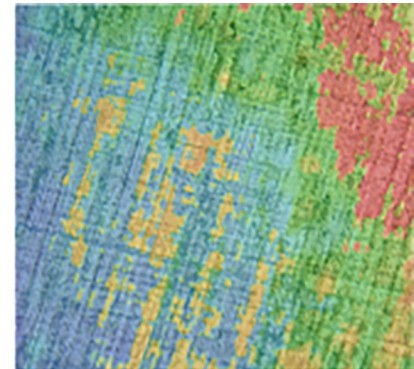

**Fig 7. Compares the optimization results.**

before optimization and the milling parameters optimized by the MSSO algorithm on the platform established above [36]. The TI64 machined surfaces were tested and analyzed, and the comparative results are shown in Figs 7 and 8 and Table 12.

Comparing the MSSO algorithm with the data of the pre-optimization variables and the objective function, it can be seen that the spindle speed is increased by 518rpm, the cutting width is reduced by 2mm, the cutting depth is increased by 0.1mm. The feed per tooth is reduced by 0.04mm, with the increase of cutting speed in high-speed cutting, the shear angle of the material in the cutting area increases, the cutting deformation coefficient decreases. The material is not deformed at high speed. The friction coefficient between the tool and the chip decreases. The actual heat generated during the cutting process decreases, and most of the heat is taken away by the chip. The temperature on the workpiece rises more slowly or even the temperature begins to decrease to reduce the work hardening of the contact surface between the tool and the titanium alloy, so the roughness of the machined surface of the titanium alloy is reduced by 55.7% compared with before optimization. It can be seen that the MSSO algorithm is used to optimize and reduce the surface roughness of titanium alloy significantly, and the MSSO algorithm optimization is reasonable and effective.

## 6 Conclusions

To obtain the better surface quality, the RSM method was used to design the TI64 milling experiment, and the regression model indicating roughness was fitted. The results of ANOVA shows that the quadratic multiple fitting coefficient ($R^2$) and the corrected multiple fitting coefficient ($R^2$) were close to 0.9, and the predicted multiple fitting coefficient ($R^2$) was close to 0.8. Therefore, the quadratic model was chosen as the analysis model of surface roughness. The relative error between the experimental value and the model predicted value is less than 3.5%, the maximum error is only 0.9%, and the average error is 0.7%, which proves that there is no significant difference between the prediction results and the experimental results.

The optimization performance of SO was improved, and the snake algorithm was optimized by orthogonal matrix initialization, adaptive parameters, and joint reverse selection

**Table 12. Before and after optimization design variables.**

| Contrast items | $X_1$(rpm) | $X_2$(mm) | $X_3$(mm) | $X_4$(mm) |
|---|---|---|---|---|
| Before optimization | 2500 | 4 | 0.2 | 0.06 |
| MSSO | 3018 | 2 | 0.3 | 0.02 |

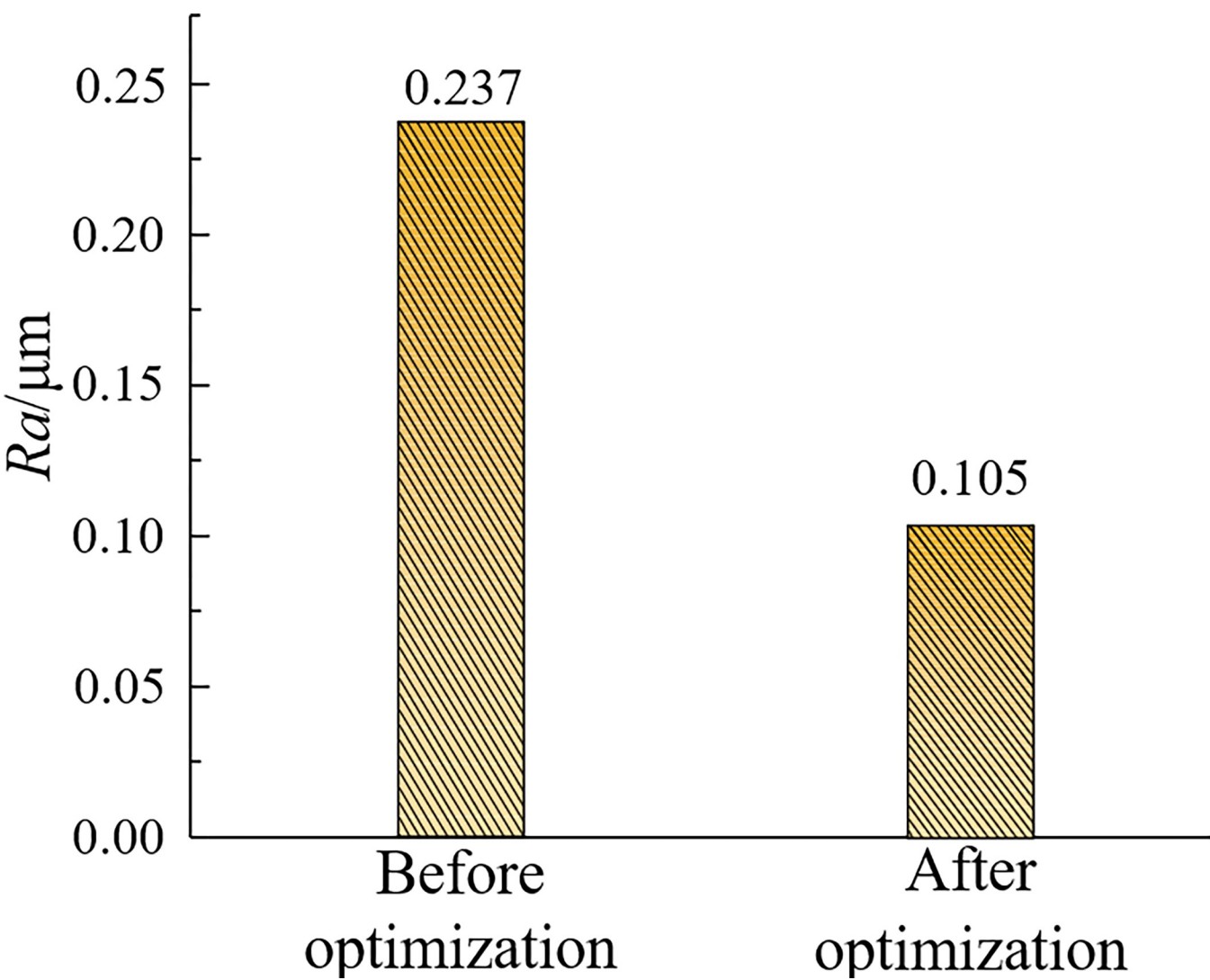

**Fig 8. Compares the cloud image of surface topography before MSSO optimization and optimization.**

strategy. The convergence, robustness, and local optimum of the MSSO algorithm were verified by the test function. In addition, the comparative optimization experiment results show that the MSSO strategies complement each other in the stage, and the algorithm is optimized from initialization to optimization and then to individual update and the defects of each stage have been significantly improved compared with the original SO.

The improved multi-strategy snake algorithm was used to optimize the milling parameters. The optimized parameters are as follows: spindle speed 3108rpm, axial cutting depth 2mm, radial cutting depth 0.3mm, feed rate 0.2mm per tooth, roughness reduction 55.7%. Compared with the surface morphology of the specimen before and after optimization, it can be seen that the optimized surface cutter row spacing is smaller, the surface height is more average, and the texture is more delicate.

## Author Contributions

**Conceptualization:** Nanqi Li, Yang Zhao, Hui Wang, Qiyuan Min.

**Data curation:** Nanqi Li, Qiyuan Min.

**Funding acquisition:** ZuEn Shang.

**Investigation:** Qiyuan Min.

**Project administration:** ZuEn Shang.

**Resources:** ZuEn Shang.

**Software:** Nanqi Li, Yang Zhao, Qiyuan Min.

**Supervision:** Hui Wang.

**Validation:** Nanqi Li, Yang Zhao.

**Visualization:** Nanqi Li.

**Writing – original draft:** Nanqi Li.

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
