## [Decision Letter · Decision Letter 0]

24 May 2024

PONE-D-24-20301Optimization of Surface Roughness for Titanium Alloy Based on Multi-strategy Fusion Snake AlgorithmPLOS ONE

Dear Dr. Shang,

Thank you for submitting your manuscript to PLOS ONE. After careful consideration, we feel that it has merit but does not fully meet PLOS ONE’s publication criteria as it currently stands. Therefore, we invite you to submit a revised version of the manuscript that addresses the points raised during the review process.

We look forward to receiving your revised manuscript.

Kind regards,

Himadri Majumder, Ph.D

Academic Editor

PLOS ONE

Journal Requirements:

2. Please note that PLOS ONE has specific guidelines on code sharing for submissions in which author-generated code underpins the findings in the manuscript. In these cases, we expect all author-generated code to be made available without restrictions upon publication of the work. 

Please review our guidelines at https://journals.plos.org/plosone/s/materials-and-software-sharing#loc-sharing-code and ensure that your code is shared in a way that follows best practice and facilitates reproducibility and reuse. 

"This study was supported by the National Natural Science Foundation of China (52204169), the project of Liaoning Provincial Science and Technology Department (2023-BS-204) and the project of Liaoning Provincial Department of Education (LJKQZ20222321), and the experimental work in this paper was supported by the Mechanical Processing Centre of Shenyang Casting Research Institute."

"This study was supported by the National Natural Science Foundation of China (52204169), the project of Liaoning Provincial Science and Technology Department (2023-BS-204) and the project of Liaoning Provincial Department of Education(LJKQZ20222321), and the experimental work in this paper was supported by the Mechanical Processing Centre of Shenyang Casting Research Institute."

Please note that funding information should not appear in the Acknowledgments section or other areas of your manuscript. We will only publish funding information present in the Funding Statement section of the online submission form. Please remove any funding-related text from the manuscript. 

6. We note that your Data Availability Statement is currently as follows: 

"All relevant data are within the manuscript and its Supporting Information files."

7. When completing the data availability statement of the submission form, you indicated that you will make your data available on acceptance. We strongly recommend all authors decide on a data sharing plan before acceptance, as the process can be lengthy and hold up publication timelines. Please note that, though access restrictions are acceptable now, your entire data will need to be made freely accessible if your manuscript is accepted for publication. This policy applies to all data except where public deposition would breach compliance with the protocol approved by your research ethics board. If you are unable to adhere to our open data policy, please kindly revise your statement to explain your reasoning and we will seek the editor's input on an exemption. Please be assured that, once you have provided your new statement, the assessment of your exemption will not hold up the peer review process.

8. Please amend the manuscript submission data (via Edit Submission) to include author "Hui Wang".

**Additional Editor Comments:**

Thank you for submitting your manuscript to PLOS ONE. I have completed my evaluation of your manuscript. The reviewers recommend reconsideration of your manuscript following revision. I invite you to resubmit your manuscript after addressing the comments.

Reviewers' comments:

Reviewer's Responses to Questions

**Comments to the Author**

1. Is the manuscript technically sound, and do the data support the conclusions?

Reviewer #1: Yes

Reviewer #2: Yes

2. Has the statistical analysis been performed appropriately and rigorously? 

Reviewer #1: Yes

Reviewer #2: Yes

3. Have the authors made all data underlying the findings in their manuscript fully available?

Reviewer #1: No

Reviewer #2: Yes

4. Is the manuscript presented in an intelligible fashion and written in standard English?

Reviewer #1: Yes

Reviewer #2: Yes

5. Review Comments to the Author

Reviewer #1: The paper introduces a milling parameter optimization method using the snake algorithm with multi-strategy fusion to enhance surface quality. The article is organized and contains distinct sections that describe the research objectives and methodology. To enhance the paper, it would be useful to include the following feedback in the revised version:

1. Some key parameters are not mentioned. The rationale on the choice of the particular set of parameters should be explained with more details. Have the authors experimented with other sets of values? What are the sensitivities of these parameters on the results?

2. A graphical abstract is needed. It should be compact and to the point.

3. The introduction paragraph should be presented more extensively; many papers were cited, but the outcomes of some of these mentioned papers on optimization were practically neglected. In addition, structure of the paper needs to revise, and all the segments should be well defined. Thus, the Authors are advised to study and incorporate few good papers on optimization as listed below and the Authors need to compare results with the recent metaheuristics if it is applicable. Moreover, the following manuscripts can be added to the list of references, which are in the domain recent algorithms: DOI: 10.32604/cmes.2023.028632; https://doi.org/10.1016/j.mex.2023.102181; https://doi.org/10.1016/j.knosys.2023.110529; DOI: 10.4018/IJAMC.2017070101

4. Programming/Codes of the proposed method should be added to reproduce/review the claimed results.

5. Authors should argue their choice of the performance evaluation indicators.

This paper can be accepted after the implementation of the suggested modification. I suggest revision for the paper.

Reviewer #2: 1. Section 1 (Introduction): It is recommended to add the following reference to the article, which is related to this study in terms of the workpiece material used in the experimental study, machining operation, experimental design and optimization, etc. https://doi.org/10.1016/j.jmapro.2022.06.016

2. How were the control factors and levels in Table 1 determined? Please explain.

3. Are the Ra results in Table 2 the arithmetic average of the surface roughness measurement results? There is a need to explain in more detail how surface roughness measurements are performed in subheading 2.1.

4. Subheading 2.1: How many times was each experiment repeated? Was a new, unworn cutting edge of the cutting tool used in each test? Please explain these issues.

5. Subheading 2.1: What is the chemical composition of Ti64? Write it as text or add it as a table.

6. Subheading 2.1: Were the experiments conducted under dry cutting conditions or with coolant? Please explain these issues. (In the photo at the bottom left of Figure 1, it can be seen that the coolant is used, but I am not sure; it cannot be seen clearly.)

7. Subheading 2.1, Paragraph 2: "……milling methods are plane milling and inverse milling." "plane millingface milling, "inverse milling" --" " up milli"g" o" "down milli"g." Please use the correct technical terms.

8. Subheading 2.1. Please specify the technical specifications of the" "APMT1604PDER-"2" insert. Write the cutting tool material. Is the cutting tool tip a cemented carbide tool, coated or uncoated tool? Why was this cutting tool material and geometry chosen for milling the workpiece material? Explain these issues. In this study, much more emphasis was placed on the statistical evaluation of experimental findings. However, to develop the machinability database of the selected workpiece and contribute to the practices in industrial applications, details of machinability experimental studies should also be included in addition to statistical evaluations.

9. Subheading 2.1: As seen in Table 1, in this experimental study, milling parameters (for example, cutting speed, feed amount, cutting depth, cutting width) were selected as input parameters. However, titanium alloy materials are among the materials that are difficult to process (for example, mill). Besides milling parameters, control factors such as cutting tool material, geometry and coating layer, and cooling environment (e.g., dry and MQL) also affect the output parameters (machinability parameters). Why were these parameters not selected as control factors apart from the selected milling parameters? Authors should also explain these issues.

10. Subheading 2.1: It has been stated that surface roughnesses were measured with a 3D microsystem. In 2D measurements using the profile meter method, Ra values are generally measured and evaluated, and the unit is a micrometer. In 3D measurements, Sa, not Ra, is measured. What is the unit of Sa measurements? Please review and analyze these evaluations.

11. Subheading 2.1: Apart from surface roughness, cutting forces, tool life, tool wear, chip type, etc. Why were the output parameters/variables not evaluated in this study? Is only evaluating the surface roughness sufficient to evaluate the machinability of titanium alloy? It is recommended that authors examine and evaluate these issues. 

12. There is a problem in the organization of the subheadings (chapters) of the article. Table 2 in Subheading 2.2 and Tables 3-5 in Subheading 2.3 should be presented under the newly created Section 3. (Experiment and experimental analysis or Result and Discussions) heading. Section 3 and Section 4 should be given as subheadings of Section 2. If the sections of the article, namely subheadings, are arranged, the article will be more organized, followable, and motivating. The information provided in these sections is very useful and necessary; only the order and organization of the subheadings are problematic, and their revision is strongly recommended.

13. Subheading 2.2: Has the procedure in RSM been applied to determine the levels of the milling parameters given in Table 1? Prove that logarithmic calculation in RSM is used to determine the levels of milling parameters. As the article's authors, please review the calculations to determine the levels of the input parameters (independent variables) in RSM.

14. Subheading 3, Subheading 4: Please refer to the explanations and information about the techniques, methods, algorithms, and equations given under these subheadings and indicate the reference numbers in the relevant texts in the article. Because many equations and methods given under these headings were not developed by the article's authors.

15. Explain the meanings of parameters, variables, constants, subscripts, and superscripts used in the equations in the article immediately after the relevant equations. Preparation of a Nomenclature for this article is highly recommended.

16. The visuality and resolution of Figure 3 should be increased. Authors can re-draw the figure in Figure 3 themselves.

17. To better understand the sequence, steps, and interrelationships of the experimental studies and statistical evaluations performed in this article, prepare a detailed flowchart and attach it to Subheading 2. It is thought that a flowchart prepared to summarize the work done in the article and reflect the steps followed will contribute greatly to the article.

18. Section 2.3: ANOVA is given in Table 4 for the predictive models developed in this study. However, in addition to the ANOVA given in Table 4, the article should also add an ANOVA reflecting the main effects of milling parameters on Ra and the effects of interactions. According to this ANOVA table to be added, the statistically effective milling parameters and their % contribution on Ra should be stated. At the same time, the effects of milling parameters on Ra should be explained in terms of machinability research by comparing similar research results in the literature. In other words, explaining the results obtained from evaluating the experimental findings with statistical methods does not provide much depth or extra contribution to the article.

6. PLOS authors have the option to publish the peer review history of their article (what does this mean?). If published, this will include your full peer review and any attached files.

Reviewer #1: No

Reviewer #2: No

---

## [Author Response · Author response to Decision Letter 0]

2 Jul 2024

Dear Dr. Majumder，

Thank you for your letter and for the reviewers’ comments concerning our manuscript entitled “Optimization of Surface Roughness for Titanium Alloy Based on Multi-strategy Fusion Snake Algorithm” (PONE-D-24-20301). Those comments are very valuable and very helpful for revising and improving our paper, as well as the important guiding significance to our researches. We have studied comments carefully and have made correction which we hope meet with approval. In the revision, we have fully addressed the comments made by the reviewers and the editor. The completed changes are as follows:

Reviewer #Editor:

Comment 1

 and https://journals.plos.org/plosone/s/file?id=ba62/PLOSOne_formatting_sample_title_authors_affiliations.pdf

Response 1

The full text format has been revised and checked in plos one style, and I would appreciate your approval.

Comment 2

Please note that PLOS ONE has specific guidelines on code sharing for submissions in which author-generated code underpins the findings in the manuscript. In these cases, we expect all author-generated code to be made available without restrictions upon publication of the work.

Please review our guidelines at https://journals.plos.org/plosone/s/materials-and-software-sharing#loc-sharing-code and ensure that your code is shared in a way that follows best practice and facilitates reproducibility and reuse.

Response 2

Ok, thanks for the suggestion. We have submitted the code for the article to the public repository：SO-MS/SO_MS.mlx at main · LiNQ-6688/SO-MS (github.com).

Comment 3

We note that the grant information you provided in the ‘Funding Information’ and ‘Financial Disclosure’ sections do not match.

Response 3

This issue has been corrected, thank you for your feedback .

Comment 4

Thank you for stating the following financial disclosure:

“This study was supported by the National Natural Science Foundation of China (52204169), the project of Liaoning Provincial Science and Technology Department (2023-BS-204) and the project of Liaoning Provincial Department of Education (LJKQZ20222321), and the experimental work in this paper was supported by the Mechanical Processing Centre of Shenyang Casting Research Institute.”

Response 4

This issue has been corrected, thank you for your feedback .

Comment 5

Thank you for stating the following in the Acknowledgments Section of your manuscript: 

"This study was supported by the National Natural Science Foundation of China (52204169), the project of Liaoning Provincial Science and Technology Department (2023-BS-204) and the project of Liaoning Provincial Department of Education(LJKQZ20222321), and the experimental work in this paper was supported by the Mechanical Processing Centre of Shenyang Casting Research Institute."

Please note that funding information should not appear in the Acknowledgments section or other areas of your manuscript. We will only publish funding information present in the Funding Statement section of the online submission form. Please remove any funding-related text from the manuscript.

Response 5

This issue has been corrected, thank you for your feedback .

Comment 6

We note that your Data Availability Statement is currently as follows:

"All relevant data are within the manuscript and its Supporting Information files."

Response 6

A change has been made in the submission system to address this issue:“Data and code are available onrequest to the authors.”

Comment 7

When completing the data availability statement of the submission form, you indicated that you will make your data available on acceptance. We strongly recommend all authors decide on a data sharing plan before acceptance, as the process can be lengthy and hold up publication timelines. Please note that, though access restrictions are acceptable now, your entire data will need to be made freely accessible if your manuscript is accepted for publication. This policy applies to all data except where public deposition would breach compliance with the protocol approved by your research ethics board. If you are unable to adhere to our open data policy, please kindly revise your statement to explain your reasoning and we will seek the editor's input on an exemption. Please be assured that, once you have provided your new statement, the assessment of your exemption will not hold up the peer review process.

Response 7

A change has been made in the submission system to address this issue:“Data and code are available onrequest to the authors.”

Comment 8

Please amend the manuscript submission data (via Edit Submission) to include author "Hui Wang".

Response 8

Changes have been made in the submission system to address this issue.

Reviewer #1:

Comment 1

 Some key parameters are not mentioned. The rationale on the choice of the particular set of parameters should be explained with more details. Have the authors experimented with other sets of values? What are the sensitivities of these parameters on the results?

Response 1

Thank you very much for your comment. When milling titanium alloys at high speed, the optimization of surface roughness is crucial as it directly affects the fatigue strength, corrosion resistance and overall performance of the part. By constructing a high-speed cutting experimental platform and applying Response Surface Methodology (RSM) and Orthogonal Test Method, we were able to systematically investigate the effects of milling parameters on surface integrity.

The experimental results show that spindle speed, width of cut, depth of cut and feed per tooth are the key factors affecting surface roughness. For example, an increase in spindle speed can reduce surface roughness to a certain extent because a higher speed reduces the time the tool is in contact with the material, which in turn reduces plastic deformation and heat accumulation. However, too high a rotational speed may lead to an increase in cutting temperature, affecting tool durability and the surface quality of the workpiece.

Adjustments to the width and depth of cut can optimize the material removal rate, which in turn affects the surface texture and roughness. A proper feed per tooth ensures stability of the cutting process and reduces surface irregularities caused by fluctuations in cutting forces. Through the combined optimization of these parameters, we aim to achieve minimum surface roughness, taking into account productivity and tool life.

In addition, by using a multi-strategy fusion snake algorithm for multi-objective optimization, we are able to improve material removal rates and achieve efficient machining while maintaining surface quality. The selection and optimization of these parameters is not only based on experimental data and statistical analysis, but also takes into account the requirements for machining efficiency and cost-effectiveness in actual production. Through these comprehensive considerations, we are able to provide a theoretical basis and data support for the manufacture of aerospace cast titanium alloy parts, ensuring the reliability of the machining process and the high performance of the parts.

For the parameter sensitivity study, the data show that the depth of cut and feed per tooth have a high sensitivity to surface roughness, while the spindle speed and cutting width, although they also have some influence, are relatively small. By considering these parameters and their interactions in a comprehensive manner, the milling process can be optimized more effectively to achieve surface roughness improvement.

However, the purpose of this paper is how to use the algorithm to select better parameters, and the significance of different parameters for the results has already been reflected in the prediction model to some extent, so it has not been described emphatically.

Comment 2

A graphical abstract is needed. It should be compact and to the point.

Response 2

We sincerely apologize for overlooking the Graphical summary, with additional graphical summary below.

Comment 3

The introduction paragraph should be presented more extensively; many papers were cited, but the outcomes of some of these mentioned papers on optimization were practically neglected. In addition, structure of the paper needs to revise, and all the segments should be well defined. Thus, the Authors are advised to study and incorporate few good papers on optimization as listed below and the Authors need to compare results with the recent metaheuristics if it is applicable. Moreover, the following manuscripts can be added to the list of references, which are in the domain recent algorithms: DOI: 10.32604/cmes.2023.028632; https://doi.org/10.1016/j.mex.2023.102181; https://doi.org/10.1016/j.knosys.2023.110529; DOI: 10.4018/IJAMC.2017070101

Response 3

We sincerely apologise for the lack of completeness of the introduction and make a request for additions.z

Comment 4

Programming/Codes of the proposed method should be added to reproduce/review the claimed results.

Response 4

Ok, thanks for the suggestion. We have submitted the code for the article to the public repository：SO-MS/SO_MS.mlx at main · LiNQ-6688/SO-MS (github.com).

Comment 5

Authors should argue their choice of the performance evaluation indicators. This paper can be accepted after the implementation of the suggested modification. I suggest revision for the paper.

Response 5

Thank you for your advice. Below we will justify the addition of the chosen performance evaluation metrics to the paper.

Surface roughness (Ra): Surface roughness is a key indicator of the surface quality of the part, which directly affects the fatigue life, wear performance and sealing performance of the part. In the high-speed milling process, the optimisation of surface roughness can significantly improve the working performance and reliability of the parts.

Reviewer #2:

Comment 1

 Section 1 (Introduction): It is recommended to add the following reference to the article, which is related to this study in terms of the workpiece material used in the experimental study, machining operation, experimental design and optimization, etc. https://doi.org/10.1016/j.jmapro.2022.06.016

Response 1

Thank you very much for your suggestion, we have included more detailed experimental procedures and testing procedures and more reference information in the paper.

Comment 2

How were the control factors and levels in Table 1 determined? Please explain.

Response 2

We agree with this suggestion and have modified the terminology throughout the text as appropriate.Regarding the selection of control factors, the control factors in Table 1 include spindle speed (X1), cutting width (X2), depth of cut (X3), and feed per tooth (X4). These factors are key parameters in milling machining, and they directly affect the machining efficiency, surface quality, tool life, and so on. In Tab 1, each control is assigned three levels, which are typically intended for Response Surface Methodology (RSM) or orthogonal experimental designs. These levels represent low, medium, and high settings of the parameter and are used to assess the effect of parameter changes on the response (e.g., surface roughness). A coding of -1, 0, and 1 is used in the table to represent the low, medium, and high levels. This coding is convenient for use in mathematical models, where 0 typically represents the medium level, and -1 and 1 represent settings below and above the medium level, respectively. In response surface analysis, these levels may be determined based on prior literature research, expert opinion, or pre-experimentation to estimate the range of possible effects of the parameters. The goal of the experimental design is to find the optimal combination of parameters to minimise or maximise a given response.The three levels of spindle speed are 500 rpm, 1500 rpm and 2500 rpm. These values are based on the capability of the machine, the recommended speed of the tool and the machining characteristics of the material. The three levels of cutting width are 2 mm, 5 mm and 8 mm, taking into account the size of the workpiece and the required material removal rate. The three levels of depth of cut are 0.1 mm, 0.25 mm and 0.4 mm, depending on the cutting capacity of the tool, the hardness of the workpiece material and the required machining accuracy. The three levels of feed per tooth are 0.01 mm, 0.03 mm and 0.04 mm, which are related to the tool geometry, the machinability of the material and the surface quality required.

Comment 3

Are the Ra results in Table 2 the arithmetic average of the surface roughness measurement results? There is a need to explain in more detail how surface roughness measurements are performed in subheading 2.1.

Response 3

We agree with this suggestion and have modified the terminology throughout the text as appropriate. we used advanced measurement techniques to evaluate the surface roughness of titanium alloy materials after high-speed milling. Specifically, a PS50 3D surface profiler was used to perform precision surface roughness measurements. This device is based on white light confocal technology, which emits a specific wavelength of light and receives the reflected light back through the probe. By calculating the distance between the probe and the surface, the device is able to accurately acquire the 3D coordinates of the surface. The collected data is analysed in detail using the NANOVIA 3D software to obtain surface roughness measurements from the nanometre to the millimetre scale. The advantages of this measurement method are its high accuracy, fast measurement and good repeatability, providing us with reliable surface roughness data. In addition, in order to further observe the macroscopic morphology of the machined surface, we also used a Japanese Keens VHX-5000 super depth-of-field microscope. This microscope provides a clear image of

---

## [Decision Letter · Decision Letter 1]

23 Jul 2024

PONE-D-24-20301R1Optimization of Surface Roughness for Titanium Alloy Based on Multi-strategy Fusion Snake AlgorithmPLOS ONE

Dear Dr. Shang,

Thank you for submitting your manuscript to PLOS ONE. After careful consideration, we feel that it has merit but does not fully meet PLOS ONE’s publication criteria as it currently stands. Therefore, we invite you to submit a revised version of the manuscript that addresses the points raised during the review process.

**ACADEMIC EDITOR: **I have now received sufficient Reviewers' feedback regarding your submitted paper. As per their feedback, I am recommending "Minor Revision" of this revised manuscript. Authors have to upgrade their paper as per the reviewer minor comments. Also read and check the whole paper for any type of technical, typo and grammatical errors. It is also advised to the authors to remove any unnecessary references, if any, which are not closely related to the present work. The reviewers suggested references are optional, authors may or may not include those in their paper.. 

We look forward to receiving your revised manuscript.

Kind regards,

Himadri Majumder, Ph.D

Academic Editor

PLOS ONE

Journal Requirements:

Additional Editor Comments:

I have now received sufficient Reviewers' feedback regarding your submitted paper. As per their feedback, I am recommending "Minor Revision" of this revised manuscript. Authors have to upgrade their paper as per the reviewer minor comments. Also read and check the whole paper for any type of technical, typo and grammatical errors. It is also advised to the authors to remove any unnecessary references, if any, which are not closely related to the present work. The reviewers suggested references are optional, authors may or may not include those in their paper.

Reviewers' comments:

Reviewer's Responses to Questions

**Comments to the Author**

1. If the authors have adequately addressed your comments raised in a previous round of review and you feel that this manuscript is now acceptable for publication, you may indicate that here to bypass the “Comments to the Author” section, enter your conflict of interest statement in the “Confidential to Editor” section, and submit your "Accept" recommendation.

Reviewer #1: All comments have been addressed

Reviewer #2: All comments have been addressed

Reviewer #3: All comments have been addressed

Reviewer #4: All comments have been addressed

2. Is the manuscript technically sound, and do the data support the conclusions?

Reviewer #1: Yes

Reviewer #2: Yes

Reviewer #3: Yes

Reviewer #4: Yes

3. Has the statistical analysis been performed appropriately and rigorously? 

Reviewer #1: Yes

Reviewer #2: Yes

Reviewer #3: Yes

Reviewer #4: Yes

4. Have the authors made all data underlying the findings in their manuscript fully available?

Reviewer #1: No

Reviewer #2: Yes

Reviewer #3: Yes

Reviewer #4: Yes

5. Is the manuscript presented in an intelligible fashion and written in standard English?

Reviewer #1: Yes

Reviewer #2: Yes

Reviewer #3: Yes

Reviewer #4: Yes

6. Review Comments to the Author

Reviewer #1: The authors have successfully addressed all my concerns raised during the review process. They have made the necessary revisions and improvements to the manuscript, ensuring that it meets the required standards. Therefore, I recommend that this paper be accepted in its current version.

Reviewer #2: The major revisions that I suggested in my previous review have been made by the authors in this revised article. Therefore, this revised article deserves to be published in your esteemed journal. My final rating for this article is "ACCEPT".

Reviewer #3: The authors addressed all the reviewers' comments. There are only some misprints that should be fixed. They are indicated in the following:

The word "we" at the beginning of a paragraph after figure 1, in the sentence "we used advanced measurement techniques…" should be capitalized since it is at the beginning of the phrase.

Please consider the same suggestion for the beginning of the phrase "the linear mode has an F-value of …" of the paragraph after table 5. In addition, there is a misprint in this paragraph related to the word "overall" at the fifth raw.

Please capitalize the first letter of the title of subsection 2.4 Regression model.

Reviewer #4: The authors corrected the paper following the recommendations. It now meets all criteria for publication. I recommend that the paper be published

7. PLOS authors have the option to publish the peer review history of their article (what does this mean?). If published, this will include your full peer review and any attached files.

Reviewer #1: **Yes: **Ghanshyam G. Tejani

Reviewer #2: No

Reviewer #3: No

Reviewer #4: **Yes: **Slobodan Tabakovic

---

## [Author Response · Author response to Decision Letter 1]

22 Aug 2024

Dear Dr. Majumder，

Thank you for your letter and for the reviewers’ comments concerning our manuscript entitled “Optimization of Surface Roughness for Titanium Alloy Based on Multi-strategy Fusion Snake Algorithm” (PONE-D-24-20301). Those comments are very valuable and very helpful for revising and improving our paper, as well as the important guiding significance to our researches. We have studied comments carefully and have made correction which we hope meet with approval. In the revision, we have fully addressed the comments made by the reviewers and the editor. The completed changes are as follows:

Comment 1

The authors addressed all the reviewers' comments. There are only some misprints that should be fixed. They are indicated in the following:

The word "we" at the beginning of a paragraph after figure 1, in the sentence "we used advanced measurement techniques…" should be capitalized since it is at the beginning of the phrase.

Please consider the same suggestion for the beginning of the phrase "the linear mode has an F-value of …" of the paragraph after table 5. In addition, there is a misprint in this paragraph related to the word "overall" at the fifth raw.

Please capitalize the first letter of the title of subsection 2.4 Regression model.

Response 1

Thank you very much for your comment. Spelling errors have been corrected and labelled in accordance with the reviewer's comments.

All authors have read and approved the re-submission of the manuscript! If you have any questions, please let me know! 

Thank you for your consideration of our paper and we are looking forward to hearing from you!

Sincerely yours,

ZuEn Shang

---

## [Editor Report · Decision Letter 2]

30 Aug 2024

Optimization of Surface Roughness for Titanium Alloy Based on Multi-strategy Fusion Snake Algorithm

PONE-D-24-20301R2

Dear Dr. Shang,

We’re pleased to inform you that your manuscript has been judged scientifically suitable for publication and will be formally accepted for publication once it meets all outstanding technical requirements.

Kind regards,

Himadri Majumder, Ph.D

Academic Editor

PLOS ONE
---

## [Editor Report · Acceptance letter]

16 Sep 2024

PONE-D-24-20301R2 

PLOS ONE

Dear Dr. Shang, 

I'm pleased to inform you that your manuscript has been deemed suitable for publication in PLOS ONE. Congratulations! Your manuscript is now being handed over to our production team.

Kind regards, 

on behalf of

Dr. Himadri Majumder 

Academic Editor

PLOS ONE